# OmniMoE: An Efficient MoE by Orchestrating Atomic Experts at Scale

**Jingze Shi** [1] **Zhangyang Peng** [1] **Yizhang Zhu** [1] **Yifan Wu** [1] **Guang Liu** [2] **Yuyu Luo** [1]

## Abstract

Mixture-of-Experts (MoE) architectures are evolving towards finer granularity to improve parameter efficiency. However, existing MoE designs face an inherent trade-off between the granularity of expert specialization and hardware execution efficiency. We propose **OmniMoE**, a system–algorithm co-designed framework that pushes expert granularity to its logical extreme. OmniMoE introduces vector-level **Atomic Experts**, enabling scalable routing and execution within a single MoE layer, while retaining a shared dense MLP branch for general-purpose processing. While this *atomic* design maximizes capacity, it poses severe challenges for routing complexity and memory access. To address these, OmniMoE adopts a system-algorithm co-design: (i) a **Cartesian Product Router** that decomposes the massive index space to reduce routing complexity from $O(N)$ to $O(\sqrt{N})$; and (ii) **Expert-Centric Scheduling** that inverts the execution order to turn scattered, memory-bound lookups into efficient dense matrix operations. Validated on seven benchmarks, OmniMoE (with 1.7B active parameters) achieves 50.9% zero-shot accuracy across seven benchmarks, outperforming coarse-grained and fine-grained baselines. Crucially, OmniMoE reduces inference latency from 73ms to 6.7ms (a $10.9\times$ speedup) compared to PEER, demonstrating that massive-scale fine-grained MoE can be fast and accurate. Our code is open-sourced at https://github.com/HKUSTDial/omni-moe.

## 1. Introduction

Mixture-of-Experts (MoEs) has emerged as a key approach to mitigating scaling bottlenecks by partially decoupling

[1]The Hong Kong University of Science and Technology (Guangzhou) [2]Beijing Academy of Artificial Intelligence. Correspondence to: Yuyu Luo <yuyuluo@hkust-gz.edu.cn>.

*Proceedings of the 43rd International Conference on Machine Learning*, Seoul, South Korea. PMLR 306, 2026. Copyright 2026 by the author(s).

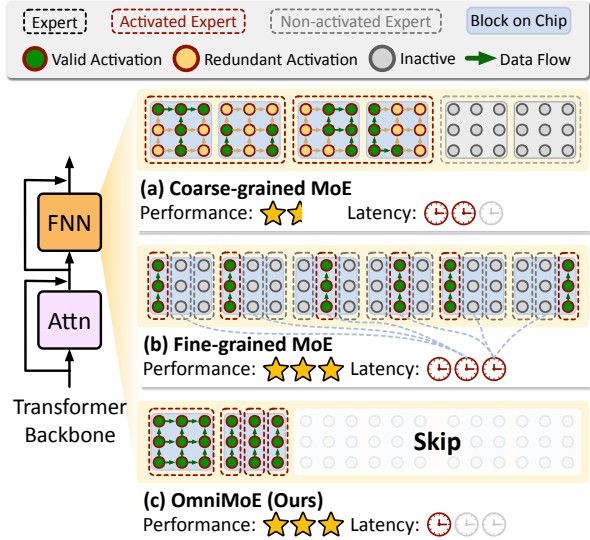

*Figure 1.* **Activation Patterns and System Optimization.** (a) Coarse-grained MoE activates large experts, inevitably involving redundant parameters and wasting computation. (b) Fine-grained MoE improves parameter efficiency, but suffers from bandwidth bottlenecks due to scattered, fragmented memory accesses. (c) Our OmniMoE employs a universally activated shared dense MLP, and uses expert-centric scheduling to reorganize fine-grained expert fetches into contiguous, coalesced memory accesses, achieving both high parameter efficiency and hardware-efficient execution.

model capacity from per-token computation (Fedus et al., 2022). By activating only a subset of experts for each token, MoEs allow for massive parameter scaling while maintaining manageable inference budgets. A central design choice in MoE is the *granularity* of experts, which largely determines both routing precision and system efficiency. Broadly, existing designs fall into two categories: coarse-grained MoEs and fine-grained MoEs.

**Coarse-Grained MoEs.** Coarse-grained architectures represent the dominant paradigm in contemporary large-scale language models. Representative systems (Du et al., 2022; Jiang et al., 2024; Zoph et al., 2022) such as DeepSeek-V3 (DeepSeek-AI et al., 2025) (256 experts) and KIMI-K2 (Team et al., 2025) (384 experts) instantiate each expert as a complete dense FFN, benefiting from hardware-efficient dense matmuls (via Tensor Cores), contiguous VRAM access, and shared-expert general knowledge and training stability (Dai et al., 2024; Nguyen et al., 2025; DeepSeek-AI et al., 2025; Team et al., 2025). Despite the success,

coarse-grained MoEs inherently suffer from *imprecise activation* (Szatkowski et al., 2024) and *low flexibility*. Specifically, activating large expert blocks incurs computation on parameters irrelevant to specific tokens (orange nodes in Figure 1 (a)), leading to computational waste (Cheng et al., 2025; Li et al., 2023; Szatkowski et al., 2024). Moreover, their rigid size hinders adaptation to limited hardware: coarse granularity restricts scaling flexibility, forcing steep, discrete memory increments when adjusting expert counts.

**Fine-grained MoEs.** Fine-grained architectures seek to maximize expressivity by utilizing millions of lightweight experts (e.g., embeddings). MoE scaling-law analyses (Ludziejewski et al., 2024; Clark et al., 2022) suggest that, under a fixed training-token budget, performance improves with the total number of activated experts. This motivates *fine-grained* MoEs (He, 2024; Nogueira dos Santos et al., 2024) that use extra lightweight experts. For example, PEER (He, 2024) scales to millions of experts by adopting a Product Key Memory (PKM (Lample et al., 2019)) style design, enabling precise routing and fine-grained control over both model capacity and activated parameters through smooth scaling.

However, scaling fine-grained experts to massive magnitudes introduces three system challenges. (i) **Limited expressivity:** existing designs (e.g., PEER (He, 2024)) reduce experts to static parameter vectors. This restricts the expert computation to linear vector aggregation, stripping away the token-dependent nonlinear transformations (e.g., MLP projections) essential for modeling complex linguistic dependencies. (ii) **Routing overhead:** scaling to a large expert pool increases routing cost and load imbalance, resulting in skewed expert utilization at scale. (iii) **Hardware inefficiency:** scattered activations trigger random memory I/O, shifting execution from compute-bound to memory-bound and degrading GPU utilization. As illustrated in Figure 1(b), while fine-grained experts ensure precise activation, the active parameters are inherently scattered across memory, which triggers frequent, non-contiguous memory accesses, inevitably shifting the execution bottleneck from computation to memory bandwidth.

While coarse-grained MoEs benefit from hardware-friendly architecture, the fine-grained ones leverage high activation efficiency and flexibility. This raises a key question: ***Is it possible to reconcile the parameter efficiency of fine-grained models with the hardware efficiency of coarse-grained architectures?*** Realizing this synergy is non-trivial. It requires a holistic orchestration that simultaneously enhances the expressivity of fine-grained experts, minimizing routing overhead in large expert spaces, and reshaping irregular sparse accesses into hardware-efficient execution.

**Our Methodology and Contributions.** To address the aforementioned challenges, we propose **OmniMoE**, a system-algorithm co-designed MoE framework that synergizes the precise parameter activation of fine-grained experts with the hardware efficiency of coarse-grained designs. The core architectural innovation lies in a hybrid parallel design that combines a shared dense MLP for capturing general semantic knowledge, with a massive pool of routed fine-grained experts that specialize in long-tail knowledge retrieval. To orchestrate the activation of these fine-grained experts at scale, we introduce three tightly integrated contributions that jointly resolve the key bottlenecks.

First, to maximize model capacity and routing precision, we push expert granularity to its logical extreme by introducing the **Atomic Expert**, a minimal routable unit parameterized by a pair of vectors, and propose a corresponding **Dynamic Expert Assembly (DEA)** mechanism to organize and compose massive experts. This formulation enables scaling to a massive expert pool and supports highly specialized, token-specific, *high-expressivity* parameter compositions.

However, orchestrating large-scale atomic experts poses an unprecedented routing challenge: standard approaches would incur prohibitive *routing overhead*. To address this, we introduce the **Cartesian Product Router**. It decomposes the massive, 1D expert index space into a two-dimensional grid. By factorizing the routing computation into two independent, low-dimensional projections, it reduces the routing cost from linear in $N$ to proportional to $\sqrt{N}$, making large-scale expert routing practical and efficient.

With routing no longer the bottleneck, the key challenge in our orchestration shifts to *hardware inefficiency*: fine-grained routing induces highly scattered atomic expert accesses, leading to poor locality and low GPU efficiency. To overcome this final obstacle, we developed **Expert-Centric Scheduling**. This systemic contribution inverts the execution paradigm from token-centric to expert-centric. By reordering computations, it groups requests targeting the same experts, thereby converting scattered memory lookups into contiguous, reusable reads and enabling the use of high-throughput Grouped GEMM operations.

In summary, OmniMoE is a system-algorithm co-designed framework that orchestrates fine-grained expert activation, from atomic expert formulation to efficient routing and hardware-aware scheduling, achieving both high model expressivity and hardware efficiency at scale. As illustrated in Figure 1, our heterogeneous architecture eliminates redundant computation inherent to coarse-grained MoEs, while our scheduling strategy resolves bandwidth bottlenecks caused by scattered, non-contiguous memory accesses under fine-grained routing. Extensive experiments demonstrate that OmniMoE achieves superior performance with **10.9× speedup** compared to state-of-the-art baselines. Our code is open-sourced at https://github.com/HKUSTDial/omni-moe.

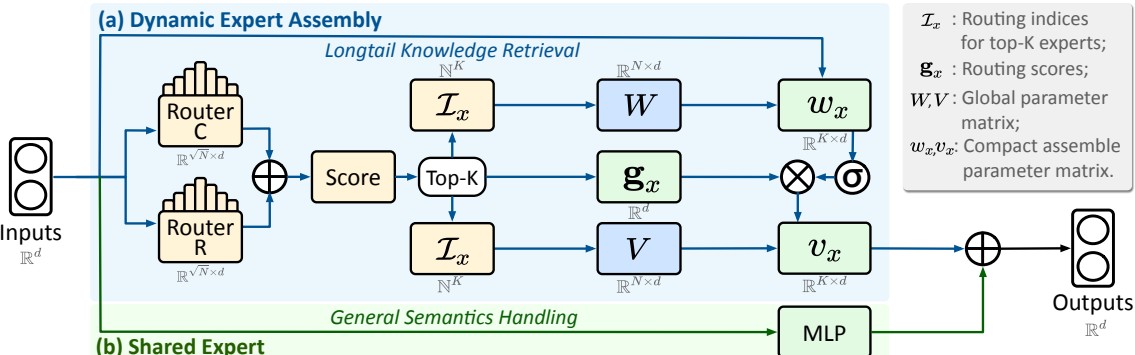

*Figure 2.* **Overview of the OmniMoE Architecture.** The framework operates via two parallel pathways to balance efficiency and expressivity. **(a) Dynamic Expert Assembly (Top):** For *Longtail Knowledge Retrieval* objective, we employ a **Cartesian Product Router** (decomposed into Row/Column routers) to efficiently compute routing scores $\mathbf{g}_x$ and identify the top-$K$ expert indices $\mathcal{I}_x$. Then the system dynamically retrieves specific parameter slices from the global matrices $W, V$ to assemble compact, token-dependent parameter blocks $w_x, v_x$ for the final gated projection. **(b) Shared Expert (Bottom):** A dense MLP which is always active to handling *General Semantics*. The final output is obtained by aggregating the outputs from the sparse, routed branch and the shared dense branch.

## 2. Methodology

We first formalize the general MoE architecture. A standard MoE layer comprises a pool of $N$ experts $\mathbb{E} = \{E_1, \ldots, E_N\}$ and a routing function $\mathcal{G}(\cdot) : \mathbb{R}^d \to \mathbb{R}^N$. Several MoE variants (Dai et al., 2024; DeepSeek-AI et al., 2025; Team, 2025) incorporate a shared dense MLP that remains universally activated for all inputs. For each input token with representation $x \in \mathbb{R}^d$, where d is the dimension of hidden states, the router computes routing scores and selects a subset of $K$ experts identified by indices $\mathcal{I}_x$.

$$\mathcal{I}_x = (I_0, \ldots, I_{K-1}) = \text{TopK}(\mathcal{G}(x), K) \quad (1)$$

where $I_i$ is the index of the $i$-th activated expert. And the routing weight $g_i$ for each expert $E_{I_i}$ could be calculated by

$$g_i = \text{Softmax}(\mathcal{G}(x)[\mathcal{I}_x])_i, i \in [0, K) \quad (2)$$

Finally, the layer output is the weighted sum of these activated experts:

$$y = \sum_{i \in [0, K)} g_i \cdot E_{I_i}(x) + \text{MLP}(x) \quad (3)$$

**OmniMoE Overview.** Figure 2 illustrates the overall architecture of OmniMoE. Our design follows the standard MoE formulation in Eq. 3. Furthermore, OmniMoE instantiates the *activated* experts as **Atomic Experts** (Section 2.1) and uses our Dynamic Expert Assembly (DEA) mechanism to retrieve and assemble token-conditioned parameters on the fly, enabling the routed branch to operate at a much finer granularity than conventional FFN experts. In parallel, we retain a dense MLP as a *shared expert* to provide general semantic reasoning and stable capacity that is independent of routing. The final representation is obtained by summing the shared dense branch and the routed fine-grained branch. The remainder of this section introduces (i) how we parameterize and store atomic experts efficiently (Section 2.1), (ii)

how we route over massive expert spaces (Section 2.2), and (iii) how we schedule the resulting sparse computations to maximize hardware efficiency (Section 2.3).

### 2.1. Atomic Experts and Dynamic Expert Assembly

In this section, we introduce the core components of our fine-grained expert design. We first define the **atomic expert** as the minimal routable computational unit. We then present **Dynamic Expert Assembly (DEA)**, our proposed mechanism for logically organizing these atomic units. Specifically, DEA enables the model to dynamically retrieve a sparse set of atomic experts from a global pool and compose them into a token-conditioned *assembled expert*.

An **atomic expert** $E_i$ is defined as a minimal, lightweight computational unit parameterized by an input vector $w_i^{in} \in \mathbb{R}^d$ and an output vector $w_i^{out} \in \mathbb{R}^d$. Given a token representation $x \in \mathbb{R}^d$, its computation is:

$$E_i(x) = \sigma(xw_i^{in\top})w_i^{out} \quad (4)$$

where $\sigma(\cdot)$ is a non-linear activation. Throughout Omni-MoE, we instantiate $\sigma(\cdot)$ with SwiGLU (Shazeer, 2020). While a single atomic expert exhibits limited expressivity, the strength of our approach arises from the dynamic composition of these experts.

The **Dynamic Expert Assembly (DEA)** mechanism governs this composition process. For each input token, DEA consists of two steps: (i) **Retrieval**, where it selects a sparse subset of the most relevant atomic experts from massive global experts, and (ii) **Assembly**, which composes the retrieved parameters into a computational block.

To make this DEA computationally feasible at scale, the parameters of all $N$ atomic experts are not stored individually. Instead, they are consolidated into two global parameter matrices, $W, V \in \mathbb{R}^{N \times d}$, which serve as a centralized pa-

rameter repository for efficient retrieval and composition.

$$W = [w_0^{in}, \ldots, w_{N-1}^{in}]^\top, \quad V = [w_0^{out}, \ldots, w_{N-1}^{out}]^\top \tag{5}$$

For a given input token $x$, the routing mechanism identifies $\mathcal{I}_x \subset \{0, ..., N-1\}$, the indices of the top-$K$ most relevant atomic experts, similar to Eq. 1. The **Retrieval** step of DEA is implemented by gathering the rows indexed by $\mathcal{I}_x$ from the global parameter matrices, yielding compact, token-local parameter blocks:

$$w_x = W[\mathcal{I}_x] \in \mathbb{R}^{K \times d}, \quad v_x = V[\mathcal{I}_x] \in \mathbb{R}^{K \times d} \tag{6}$$

Simultaneously, the associated routing scores are collected into a vector $\mathbf{g}_x = [g_{I_0}, \ldots, g_{I_{K-1}}] \in \mathbb{R}^K$. The **Assembly** step is then performed by composing these retrieved parameters and their corresponding weights into a single, fused computation:

$$y = (\mathbf{g}_x \odot \sigma(xw_x^\top))v_x + \text{MLP}(x). \tag{7}$$

This formulation demonstrates how DEA effectively constructs a unique, powerful assembled expert for each token by composing simple, reusable atomic experts. This approach ensures that every retrieved parameter is computationally active for the target token, achieving extreme parameter efficiency while maintaining high expressivity.

## 2.2. Cartesian Product Router

The primary role of the router is to efficiently select a sparse set of atomic expert indices $\mathcal{I}$ for the DEA mechanism. However, scaling to massive expert pools presents a severe indexing challenge. A standard top-$K$ router computes routing scores for all experts via a projection matrix $W_g \in \mathbb{R}^{d \times N}$ and set $\mathcal{G}(x) = xW_g$ in Eq. 2. When $N$ reaches millions, the computational cost of $\mathcal{G}(x)$ ($O(Nd)$) and the memory required to store $W_g$ becomes prohibitively expensive and often dominates the total inference latency.

**Intuition.** The key insight is that the one-dimensional expert index space of size $N$ can be decomposed into the Cartesian product of two lower-dimensional subspaces. Therefore, to overcome this bottleneck, we introduce the **Cartesian Product Router**. Rather than scoring all $N$ experts with a single $N$-way classifier, we view an expert id $n$ as a *2D coordinate* $(i, j)$ on a $N_r \times N_c$ grid (with $N = N_r N_c$). The router predicts two low-dimensional distributions over *rows* and *columns*, and composes them to score any expert on the grid. This is analogous in spirit to product-structured indexing (e.g., PKM (Lample et al., 2019)): We replace one prohibitively large projection with two small projections, while still addressing the full $N$-sized expert space.

**Implicit Scoring via Factorized Projections.** Our modeling assumption is that the joint probability distribution over

the expert grid can be approximated by the product of two independent marginal distributions:

$$p(i, j|x) \approx p_r(i|x) \cdot p_c(j|x). \tag{8}$$

We replace the single routing matrix $W_g$ with two smaller matrices, $W_r \in \mathbb{R}^{d \times N_r}$ and $W_c \in \mathbb{R}^{d \times N_c}$. For an input token $x$, the row and column logits are computed as

$$s_r = xW_r, \; s_c = xW_c. \tag{9}$$

Then the log-probabilities for each subspace are obtained via the LogSoftmax function for numerical stability:

$$p_r = \text{LogSoftmax}(s_r), \; p_c = \text{LogSoftmax}(s_c). \tag{10}$$

Since $p_r$ and $p_c$ are *log*-probabilities (via LogSoftmax), the factorized product becomes additive in log-space: $(\log p(i, j \mid x) \approx p_r[i] + p_c[j])$. The score for an expert at coordinate $(i, j)$ is the sum of the corresponding log-probabilities, which implicitly defines a score matrix $\mathcal{S} \in \mathbb{R}^{N_r \times N_c}$ without its materialization:

$$\mathcal{S}_{ij} = p_r[i] + p_c[j] \tag{11}$$

**Parallel Top-$K$ Selection.** Although $\mathcal{S} \in \mathbb{R}^{N_r \times N_c}$ is never materialized, its entries can be computed on-the-fly as described in Eq. 11 from the global vectors $p_r$ and $p_c$. We therefore partition the implicit grid into tiles and assign each tile to parallel GPU thread blocks. Each block computes scores for its tile and extracts local top-$K$ candidates. These candidates are then merged via a lightweight reduction to obtain the global top-$K$ expert indices $\mathcal{I}_x$. The routing weights $\mathbf{g}_x$ are computed by normalizing the corresponding top-$K$ scores according to Eq. 2.

**Complexity Analysis.** The factorized router reduces the projection cost (and router parameter size) from $O(Nd)$ to $O(\sqrt{N}d)$. Top-$K$ selection is performed on the implicit grid via a tiled GPU search and reduction; although the total score-evaluation work still scales with $N$, it is highly parallel and incurs negligible wall-clock overhead in practice. See Appendix B for the full derivation and details.

## 2.3. Expert-Centric Scheduling

The Cartesian Product Router (Section 2.2) operates on a per-token basis: for each input token $x$, it efficiently selects the top-$K$ expert indices $\mathcal{I}_x$ and computes gating weights $\mathbf{g_x}$. In practice, however, execution processes a *batch* of tokens $\mathcal{X} = \{x_l\}_{l=0}^{L-1}$. While routing is efficient, token-centric execution becomes a bottleneck at batch scale: each token independently fetches its selected expert parameters from HBM via scattered accesses, fragmenting memory traffic and preventing vectorization, thus limiting throughput.

To address this, we propose **Expert-Centric Scheduling**. Unlike standard static approaches that iterate over all experts, our method dynamically organizes computation based

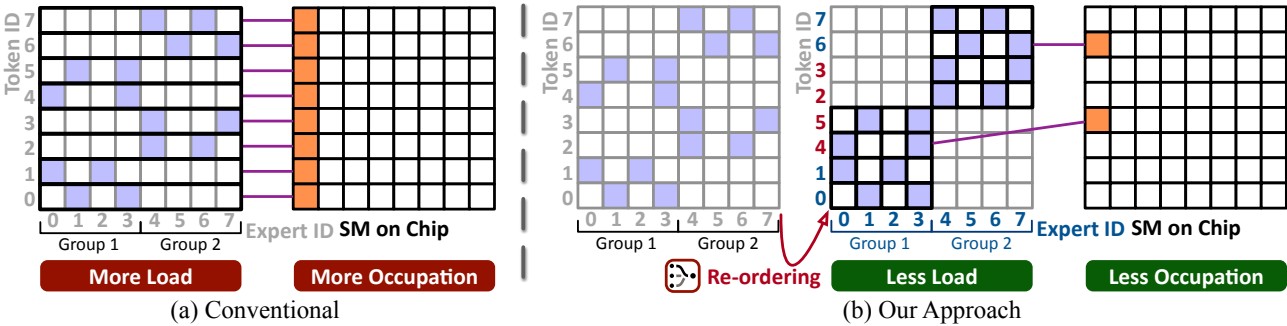

*Figure 3.* **Comparison of Execution Paradigms: Token-Centric vs. Expert-Centric Scheduling. (a) Conventional:** Tokens independently fetch parameters from scattered experts, leading to random memory accesses (high load overhead) and fragmented vector-vector computations that underutilize on-chip SMs. **(b) Our Approach:** We invert the execution order using expert-centric scheduling. **Left-to-Right:** First, tasks are reordered: we compress active experts into dense groups (e.g., experts 0–3 are grouped into Group 1) and sort tasks by Token ID within each group. **Matrix Fusion:** This reorganization allows us to merge individual token-expert pairs into dense tensors. Instead of scattered ops, the GPU executes efficient **Grouped GEMM** kernels (rightmost block), where a block of expert weights is loaded once and reused across stacked tokens, maximizing Tensor Core utilization and memory bandwidth.

on *active* experts. The pipeline proceeds as follows: we first *collect* routed computation tasks, *group* physically nearby experts, then *reorder* tasks according to these groups to improve locality, and finally *execute* the resulting workloads using high-throughput GEMM kernels.

**Task Collection and Active Expert Compression.** For a batch of $L$ tokens $\mathcal{X} = \{x_l\}_{l=0}^{L-1}$ with top-$K$ routing, the router returns, for each token $l$, an ordered expert index list $\mathcal{I}_l = (I_{l,0}, \dots, I_{l,K-1})$ and the corresponding gating weights $\mathbf{g}_l = (g_{l,0}, \dots, g_{l,K-1})$, where $g_{l,k}$ denotes the routing weight assigned to expert $I_{l,k}$. We first flatten these decisions into a list of $M = L \times K$ tasks:

$$\tilde{\mathcal{T}} = \left\{ (x_l, I_{l,k}, g_{l,k}) \mid l \in [0, L), \ k \in [0, K) \right\} \quad (12)$$

and collect the set of **unique active experts** $\mathbb{E}_{\text{active}} = \bigcup_{x \in \mathcal{X}} \mathcal{I}_x$, sorted by global expert ID. We then partition this ordered list into **contiguous groups** of size $B$, so that experts with nearby IDs are placed in the same group. Specifically, the $\tau$-th expert in $\mathbb{E}_{\text{active}}$ is assigned to group $q_\tau = \lfloor \tau/B \rfloor$. Consequently, the number of execution groups is determined solely by the active sparsity, i.e., $N_{\text{groups}} = \lceil |\mathbb{E}_{active}|/B \rceil$, ensuring that every compute group (except the last) is fully populated.

**Hierarchical Sorting.** We reorganize the tasks $\tilde{\mathcal{T}}$ by performing a hierarchical sort. The primary key is the Group ID $q$, and the secondary key is the Token ID $l$.

$$\mathcal{T} = \text{Sort}(\tilde{\mathcal{T}}, \text{keys} = (q, l)) \quad (13)$$

This sorting strategy improves hardware efficiency at two levels: (i) *Inter-Group Locality:* tasks targeting the same set of $B$ active experts are clustered together; (ii) *Intra-Group Coalescing:* within each group, processing tasks in increasing order of token ID $l$ improves coalescing for input reads and output scatters.

**Grouped GEMM Execution.** After hierarchically sorting the tasks, we process each of the $N_{groups}$ active groups. For a given group $q$, we first gather the corresponding expert parameters into dense blocks $W_q, V_q \in \mathbb{R}^{B \times d}$. Concurrently, we stack the input tokens and gating weights for all tasks assigned to this group, forming a dense input tensor $\mathbf{X}_q \in \mathbb{R}^{T_q \times d}$ and a gating vector $\mathbf{G}_q \in \mathbb{R}^{T_q \times B}$, where $T_q$ denotes the number of tasks in group $q$. The entire computation is then performed by a single fused operation:

$$\mathbf{O}_q = (\mathbf{G}_q \odot \sigma(\mathbf{X}_q W_q^\top)) V_q \quad (14)$$

where $\mathbf{O}_q \in \mathbb{R}^{T_q \times d}$ is the output block. The per-task outputs are subsequently written back via scatter-add, preserving the semantics of Eq. 7.

**Why Expert-Centric Scheduling is Efficient.** As illustrated in Figure 3, unlike the token-centric paradigm, our expert-centric approach clusters spatially proximate experts into contiguous groups, reorders tasks by token ID within each group, and fuses the resulting workloads into a small number of high-throughput Grouped GEMM kernels. By executing tasks in expert-centric order and sorting by token ID within each group, we increase parameter reuse and improve memory locality, which raises effective bandwidth and enables high-throughput Grouped GEMM execution. A detailed complexity analysis is deferred to Appendix A.

## 3. Experiments

### 3.1. Experimental Setup

**Model Architectures.** We compare six FFN variants: (i) **Dense** (standard MLP), (ii) **Gshard** (Lepikhin et al., 2021), (iii) **DeepSeekMoE** (Dai et al., 2024), (iv) **PKM** (Lample et al., 2019), (v) **PEER** (He, 2024), and (vi) **OmniMoE** (ours). All models adopt Grouped Query Attention (GQA) (Ainslie et al., 2023). For fair comparison, we keep

the Transformer backbone identical across methods (depth, width, and attention configuration) and vary only the FFN module. It is important to emphasize that we prioritize **architectural comparison** via controlled **pre-training from scratch** rather than comparing against off-the-shelf checkpoints. We define the activated-parameter budget as the number of unique parameters utilized in the forward pass of a single token, including embeddings, attention weights, the shared dense FFN, router projections, and the top-$K$ active MoE experts. For efficiency baselines, we ensure state-of-the-art implementations (see Appendix B for details).

We evaluate OmniMoE under three complementary settings. For **Speed and Memory Benchmarking**, we fix a **200M** backbone and sweep the activated-parameter budget (Act Params) and the number of activated tokens (Act Tokens) as summarized in Table A (see Appendix B). For **scaling-law** experiments, we train MoE families with **280M-A80M, 800M-A200M, 2.7B-A680M, and 6.4B-A1.7B** configurations (where A denotes the activated parameter budget) alongside their **Dense** counterparts with matched activated parameters, using matched backbones and training recipes across baselines (Table B in Appendix B). For **downstream evaluation**, we report zero-shot results using the **6.4B-A1.7B** models.

**Training Data and Tokenization.** Models are pre-trained on the SmolLMCorpus (Allal et al., 2025), a high-quality corpus of 40 billion tokens spanning Web, Textbook, Code, and Math domains. This diverse mixture establishes fundamental linguistic proficiency and broad general knowledge. We employ the NeoX tokenizer (Black et al., 2022) with a vocabulary size of 128,256 tokens.

**Training Strategy and Hyper-Parameters.** We use the AdamW optimizer (Loshchilov & Hutter, 2017) with the WSD learning rate scheduler (Hägele et al., 2024). Hyperparameters follow optimal scaling laws (Li et al., 2025) and Chinchilla compute-optimality protocols (Hoffmann et al., 2022). We run experiments in the NVIDIA PyTorch container (NVIDIA, 2022) with Hugging Face Transformers (Wolf et al., 2020). All inference/evaluation runs use a single node with $8\times$ NVIDIA A100 GPUs.

**Evaluation Benchmarks.** We evaluate downstream and reasoning performance with Hugging Face LightEval (Fourrier et al., 2023) on seven commonsense benchmarks: MMLU (Hendrycks et al., 2021a) (multitask knowledge), TriviaQA (Joshi et al., 2017) (factual recall), ARC (Clark et al., 2018) (science reasoning), PIQA (Bisk et al., 2020) (physical commonsense), HellaSwag (Zellers et al., 2019) (commonsense inference), OBQA (Mihaylov et al., 2018) (open-book QA), and Winogrande (Sakaguchi et al., 2019) (coreference resolution); and five extended benchmarks: GSM8K (Cobbe et al., 2021) (math reasoning), MATH (Hendrycks et al., 2021b) (competition math),

BBH (Suzgun et al., 2022) (hard reasoning), MBPP (Austin et al., 2021) (code generation), and HumanEval (Chen et al., 2021) (code synthesis).

## 3.2. Main Results

**Main Results.** We summarize the main empirical findings of OmniMoE from two complementary perspectives: *model quality* under a fixed active-parameter budget, and *system efficiency* (latency/memory) when executing the corresponding routed feed-forward computation.

**Downstream Performance.** As shown in Table 1, our 6.4B-A1.7B model achieves the best average zero-shot accuracy (50.9), outperforming both coarse-grained (e.g., +0.7 vs. DeepSeekMoE) and fine-grained (+2.0 vs. PEER) baselines. The results highlight the benefits of our heterogeneous design: compared to coarse-grained models, OmniMoE's precision on knowledge-intensive tasks like TriviaQA (+1.1) and OBQA (+1.4) is superior. Conversely, compared to fine-grained models with limited expressivity, OmniMoE's shared dense expert boosts performance on reasoning-heavy benchmarks such as ARC (+3.6) and HellaSwag (+4.6).

**Reasoning Performance and Throughput.** To further validate generalization beyond commonsense tasks, we report results on reasoning benchmarks alongside end-to-end throughput. As shown in Table 2, OmniMoE achieves the highest throughput among all MoE variants ($\sim$14.2k tok/s), comparable to the Dense baseline ($\sim$14.8k tok/s) and significantly outperforming fine-grained baselines (PEER: $\sim$11.6k tok/s, PKM: $\sim$7.9k tok/s). This confirms that Expert-Centric Scheduling effectively eliminates the memory bandwidth bottleneck at the full-model level. On the extended benchmarks, OmniMoE achieves an average score of 42.5, outperforming DeepSeekMoE (40.5) by +2.0 and PEER (27.0) by +15.5. The large gap over PEER indicates that purely fine-grained designs without a shared dense backbone struggle with sustained multi-step logic, while OmniMoE's hybrid architecture effectively combines precise knowledge retrieval with stable reasoning capacity.

**End-to-End Efficiency and Scalability.** Figure 4 demonstrates that OmniMoE is substantially more efficient. Despite activating a comparable or even larger number of parameters (OmniMoE: 28M, PEER: 26M, DeepSeekMoE: 28M), OmniMoE achieves substantially lower latency, reducing inference time from 73 ms (PEER) and 102 ms (DeepSeekMoE) to 6.7 ms at 4,096 tokens, **10.9$\times$ and 15.2$\times$ speedup** respectively, while maintaining a memory footprint comparable to coarse-grained MoEs. This gain stems directly from Expert-Centric Scheduling, which transforms scattered memory accesses into coalesced, reusable reads, thus shifting the execution from memory-bound to compute-bound.

*Table 1.* **Performance on Downstream Benchmarks for 6.4B-A1.7B MoE Models and the 1.7B Dense Baseline..** The best results for each size are in bold, and the second-best results are underlined. For the pre-trained base model, OmniMoE performs well on most tasks, demonstrating its effectiveness.

| MODEL | MMLU | TRIVIAQA | ARC | PIQA | HELLASWAG | OBQA | WINOGRANDE | AVG. |
|---|---|---|---|---|---|---|---|---|
| | ACC ↑ | ACC ↑ | ACC ↑ | ACC ↑ | ACC ↑ | ACC ↑ | ACC ↑ | AVG ↑ |
| Dense | 35.4 | 9.4 | 53.4 | 72.9 | 56.1 | 37.0 | 57.3 | 45.9 |
| Gshard | 36.7 | 16.7 | 58.3 | 75.3 | 59.3 | 38.7 | 59.5 | 49.2 |
| DeepSeekMoE | 37.1 | 17.4 | 60.7 | 77.2 | **61.2** | 38.9 | 59.1 | 50.2 |
| PKM | 36.3 | 12.2 | 53.6 | 73.8 | 52.7 | 38.2 | 56.7 | 46.2 |
| PEER | 37.4 | 16.9 | 57.4 | 75.9 | 56.3 | 39.1 | 59.4 | 48.9 |
| OmniMoE (ours) | **37.5** | **18.5** | **61.0** | **78.7** | 60.9 | **40.3** | **59.7** | **50.9** |

*Table 2.* **Performance and Throughput on Reasoning Benchmarks for 6.4B-A1.7B MoE Models and the 1.7B Dense Baseline.** Throughput is measured with 40 batched parallel requests (∼512 input tokens, ∼8192 output tokens) on 8× A100 GPUs. The best results are in bold, and the second-best are underlined. For the fine-tuned models, OmniMoE performs well on most tasks and achieves the highest throughput.

| MODEL | THROUGHPUT | MMLU | BBH | GSM8K | MATH | ARC-C | MBPP | HUMANEVAL | AVG. |
|---|---|---|---|---|---|---|---|---|---|
| | (tok/s) | ACC ↑ | ACC ↑ | ACC ↑ | ACC ↑ | ACC ↑ | ACC ↑ | ACC ↑ | ↑ |
| Dense | ∼14.8k | 46.4 | 37.7 | 46.3 | 11.7 | 37.4 | 40.0 | 6.7 | 32.3 |
| Gshard | ∼14.0k | 48.2 | 41.5 | 61.9 | 18.9 | 44.1 | 47.4 | 8.5 | 38.6 |
| DeepSeekMoE | ∼13.4k | 49.1 | 43.0 | 65.6 | 21.3 | 45.6 | 49.9 | 9.1 | 40.5 |
| PKM | ∼7.9k | 36.3 | 24.8 | 18.9 | 4.1 | 30.2 | 17.6 | 1.8 | 19.1 |
| PEER | ∼11.6k | 43.8 | 35.6 | 31.4 | 7.9 | 35.9 | 29.7 | 4.9 | 27.0 |
| OmniMoE (ours) | ∼14.2k | **50.2** | **44.6** | **69.8** | **24.1** | **46.8** | **52.3** | **9.8** | **42.5** |

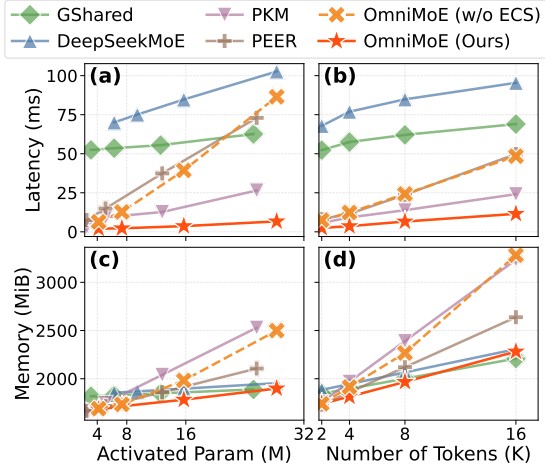

*Figure 4.* **End-to-End Efficiency Comparison.** (a, b) Inference latency and (c, d) peak memory versus activated parameters (left column) and input token count (right column). Baselines include Dense, Gshard, DeepSeekMoE, PKM, and PEER. OmniMoE achieves consistently lower latency than DeepSeekMoE and fine-grained baselines (PKM/PEER), while maintaining a peak memory footprint comparable to coarse-grained MoEs.

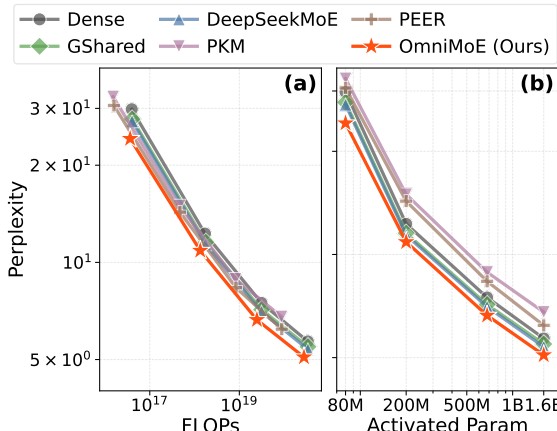

*Figure 5.* **Scaling Laws.** Validation perplexity (lower is better) versus (a) training FLOPs and (b) activated parameters. OmniMoE consistently outperforms all baselines, achieving the best trade-off between model quality and computational cost.

Interestingly, although DeepSeekMoE uses coarse-grained FFN experts, it can be slower than fine-grained PEER at large token counts or activated budgets. This is largely due to packing/alignment overhead in tiled coarse-grained kernels, where routed tokens must be reordered and

padded to fixed block sizes, causing redundant computation and extra memory traffic. In contrast, OmniMoE reshapes fine-grained activations into compact, expert-centric batched matrix operations. We report strict end-to-end latency for all methods, including all scheduling/reordering overheads for OmniMoE and the corresponding layout-transformation/alignment costs for baselines, confirming net gains from improved hardware utilization. Furthermore, we verify the scalability of OmniMoE in distributed training

settings (Appendix C). We observe that the communication overhead saturates once the expert pool size exceeds the active token count, demonstrating that OmniMoE can scale to millions of experts with constant communication cost.

**Scaling laws (perplexity vs. compute/activated parameters).** Figure 5 compares the scaling behavior of different FFN variants. Under matched training FLOPs, OmniMoE consistently achieves the lowest perplexity among all baselines, indicating superior compute efficiency. Moreover, when controlling for the activated parameter budget, OmniMoE also attains the lowest perplexity, demonstrating higher parameter efficiency. As the activated capacity increases, OmniMoE benefits more steadily from additional experts, reflecting the complementary roles of fine-grained activation for long-tail knowledge and the shared dense MLP branch for stable general reasoning.

### 3.3. Ablation Studies

We ablate the three core components of OmniMoE to isolate their individual contributions. Table 3 isolates the impact of three core components in OmniMoE: (i) the **Shared Dense MLP** for general stability, (ii) the **Cartesian Product Router** for routing quality, and (iii) **Expert-Centric Scheduling** for system efficiency. All metrics are normalized to the full model (lower is better for Latency/Memory/PPL/Unevenness). To characterize expert utilization, we follow PKM (Lample et al., 2019) and PEER (He, 2024) and report two distribution metrics based on the normalized expert retrieval frequency $z \in \mathbb{R}^N$:

- **Expert Usage**: The fraction of experts activated at least once, defined as $\frac{1}{N}|\{i \mid z_i > 0\}|$.

- **Unevenness**: The KL divergence from a uniform distribution, computed as $D_{\text{KL}}(z\|\mathcal{U}) = \sum_i z_i \log(N z_i)$, where lower values indicate more balanced load.

**Effect of the shared dense MLP.** Removing the shared dense MLP slightly improves efficiency ($0.86\times$ latency, $0.98\times$ memory) but hurts both perplexity ($1.2\times$) and downstream performance ($0.91\times$ knowledge and $0.79\times$ reasoning). This suggests that the shared dense branch serves as a critical foundational backbone complementary to fine-grained retrieval. It handles common linguistic patterns and reasoning steps, allowing the routed branch to focus exclusively on fetching token-specific long-tail knowledge.

**Effect of the Cartesian Product Router.** Replacing the Cartesian Product Router with a standard dense routing projection leads to a dramatic efficiency regression ($30.6\times$ latency and $337.5\times$ memory), driven by the cost of computing and storing full-dimension logits. Crucially, it also degrades model quality ($1.4\times$ PPL). Notably, *expert usage* collapses to only 4% and unevenness increases from 0.24 to

The negative coefficient on education should not be interpreted causally because it likely reflects omitted variable bias or multicollinearity rather than a true negative effect. The high variance inflation factors indicate severe multicollinearity, which distorts coefficient estimates and makes causal claims unreliable.

Omitted variable bias arises when unobserved factors correlated with both education and wages are excluded, potentially reversing the sign of the coefficient. Multicollinearity, evident here, inflates standard errors and can cause coefficient sign reversals, even when the true effect is positive. Suppression effects occur when adding variables reveals a negative relationship masked by confounding, but multicollinearity complicates this interpretation.

A high $R^2$ indicates the model fits the data well for prediction but does not imply causality, as it captures correlations rather than causal mechanisms. The model's improved fit simply means the added variables explain additional variance in wages, not that the negative education coefficient is valid.

The most defensible interpretation is that the negative coefficient is a statistical artifact due to multicollinearity and omitted variables, not evidence of a true negative causal effect. Researchers should address multicollinearity and consider alternative models or data to isolate education's effect more reliably.

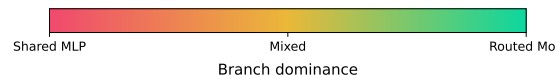

Shared MLP     Mixed     Routed MoE

Branch dominance

*Figure 6.* **Token-Level Branch Dominance Analysis.** Each token is colored by which branch contributes more to the final prediction: the Shared MLP (red) or the Routed MoE (green). Factual entities and rare tokens are predominantly served by the routed atomic experts, while common linguistic patterns and reasoning connectives are handled by the shared dense MLP, empirically validating the functional specialization of the two branches.

0.77. Despite rigorous tuning of the standard auxiliary load-balancing loss for this baseline during training, the naive gate fails to learn distinct specializations over the massive expert space, collapsing into a few dominant experts.

**Effect of Expert-Centric Scheduling.** Reverting Expert-Centric Scheduling to the standard token-centric execution mechanism preserves quality (metrics remain at $1.0\times$) but incurs a massive system cost ($24.8\times$ latency and $417.7\times$ memory). The peak memory increase corresponds to the materialization of full routing tensors required by the standard baseline, which our scheduling avoids. This confirms that our scheduling strategy is the primary source of acceleration: by inverting the loop order to process experts sequentially, we transform strictly random HBM accesses into streaming reads and maximize on-chip tensor reuse, eliminating the memory bandwidth bottleneck inherent to fine-grained MoEs.

**Branch Specialization Analysis.** To directly validate the

*Table 3.* **Ablation Study**. All metrics are reported relative to the full model. Lower is better for Latency, Memory, PPL, and Unevenness; higher is better for Knowledge Performance, Reasoning Performance, and Expert Usage.

| METHODS | LATENCY ↓ | MEMORY ↓ | PPL ↓ | KNOWLEDGE PERF. ↑ | REASONING PERF. ↑ | EXPERT USAGE ↑ | UNEVENNESS ↓ |
|---|---|---|---|---|---|---|---|
| Full | 1.0× | 1.0× | 1.0× | 1.0× | 1.0× | 100% | 0.24 |
| w/o Shared Dense MLP | 0.86× | 0.98× | 1.2× | 0.91× | 0.79× | 100% | 0.27 |
| w/o Cartesian Product Router | 30.6× | 337.5× | 1.4× | 0.66× | 0.79× | 4% | 0.77 |
| w/o Expert-Centric Scheduling | 24.8× | 417.7× | 1.0× | 1.0× | 1.0× | 100% | 0.24 |

division of labor between the two branches, we visualize per-token branch dominance in Figure 6. For each token, we measure the relative contribution of the shared dense MLP versus the routed atomic experts to the final hidden state. The results reveal a clear functional specialization: factual entities and domain-specific terms are predominantly served by the routed branch, while common function words and reasoning connectives rely on the shared MLP. This pattern is consistent with the ablation findings in Table 3, where removing the shared MLP disproportionately hurts reasoning performance while preserving expert usage, confirming that the two branches fulfill complementary roles.

## 4. Related Work

**MoE Architectures.** Conditional computation via Mixture-of-Experts (MoE) enables scaling model capacity with bounded per-token cost (Shazeer et al., 2017; Sun et al., 2025; Mu & Lin, 2025; Liu et al., 2025). Early MoE Transformers predominantly adopt coarse-grained FFN experts with lightweight routing, exemplified by Switch Transformers' Top-1 gating (Fedus et al., 2022) and GShard (Lepikhin et al., 2021). Motivated by scaling-law evidence for improved specialization, recent work trends toward finer-grained expert designs (Ludziejewski et al., 2024; Tian et al., 2025); modern LLMs (e.g., DeepSeek-V3 (DeepSeek-AI et al., 2025), KIMI-K2 (Team et al., 2025)) scale to hundreds of experts and often include shared experts to stabilize general knowledge (Dai et al., 2024). Pushing granularity to the extreme, PKM (Lample et al., 2019) and PEER (He, 2024) replace FFNs with million-scale tiny experts (e.g., embeddings), improving routing precision but reducing per-expert expressivity. Additionally, prior work reports redundant activation in FFNs/MoEs (Li et al., 2023; Szatkowski et al., 2024; Zhou et al., 2025; Yang et al., 2024); MoNE (Cheng et al., 2025) prunes computation within activated coarse-grained experts but retains expert-level top-$K$ routing, whereas OmniMoE routes over atomic experts to enable finer-grained control of activated parameters.

**Efficient MoE Systems.** System optimizations for MoE generally focus on kernel fusion and communication scheduling for coarse-grained experts. Frameworks such as DeepSpeed-MoE (Rajbhandari et al., 2022), Fast-MoE (He et al., 2021), and MegaBlocks (Gale et al., 2023) opti-

mize GEMM kernels and handle variable-length sequences to mitigate padding overheads in coarse-grained MoEs, whereas our Expert-Centric Scheduling targets fine-grained atomic experts, converting scattered memory accesses into contiguous batched operations. Recent work like Sonic-MoE (Guo et al., 2025) improves efficiency with memory-efficient algorithms, minimal activation caching, and tile-aware token rounding to reduce padding waste in Grouped GEMM kernels. Other works, including PIT (Zheng et al., 2023) and ScatterMoE (Tan et al., 2024), further exploit dynamic sparsity to prune invalid computations within activated coarse-grained experts but retain expert-level top-$K$ routing. In contrast, OmniMoE introduces Expert-Centric Scheduling to transform scattered memory accesses into hardware-efficient batched operations. Concurrent works Klotski (Fang et al., 2025) and ExpertFlow (He et al., 2026) also batch tokens per expert but target inter-request scheduling and cross-device load balancing for deployed coarse-grained MoEs; our scheduling instead operates intra-batch on a single device, addressing the scattered HBM reads unique to million-scale atomic experts.

## 5. Conclusion

In this paper, we presented **OmniMoE**, a system-algorithm co-designed MoE framework that integrates a shared dense MLP for general-purpose reasoning with massive atomic experts for long-tail knowledge retrieval, thereby enabling more precise parameter activation. To make large-scale expert activation practical, OmniMoE orchestrates the activation of atomic experts via two key innovations: the **Cartesian Product Router** and **Expert-Centric Scheduling**. Together, these components yield a dramatic $10.9\times$ inference speedup over the state-of-the-art fine-grained baseline, PEER. Moreover, under comparable activated-parameter budgets, OmniMoE consistently improves average accuracy and outperforms strong baselines on most benchmarks. These results demonstrate that, with holistic co-design, massive-scale fine-grained MoEs can be both accurate and highly efficient. Our current implementation relies on custom Triton kernels optimized for NVIDIA GPUs; portability to other hardware backends remains unexplored. Additionally, scaling behavior beyond 6.4B total parameters has not yet been validated empirically and is left for future work.

## Acknowledgements

This paper was supported by the NSF of China (62402409); Youth S&T Talent Support Programme of Guangdong Provincial Association for Science and Technology (SKXRC2025461); the Young Talent Support Project of Guangzhou Association for Science and Technology (QT-2025-001); Guangzhou Basic and Applied Basic Research Foundation (2026A1515010269, 2025A04J3935, 2023A1515110545); and Guangzhou-HKUST(GZ) Joint Funding Program (2025A03J3714). We gratefully acknowledge the FlagOS open-source community and the OpenSeek project for providing the computational resources essential to this work.

## Impact Statement

This paper presents work whose goal is to advance the field of Machine Learning. There are many potential societal consequences of our work, none which we feel must be specifically highlighted here.

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

# A. Complexity Analysis

In this section, we provide a detailed complexity derivation for both the Cartesian Product Router and the Expert-Centric Scheduling strategy. We focus on theoretical FLOPs for routing and Memory Traffic (I/O) for scheduling, as these correspond to the primary bottlenecks in each stage.

**Routing Complexity Derivation.** Consider a standard Top-$K$ router with $N$ experts and hidden dimension $d$. For a single token $x \in \mathbb{R}^d$, the router computes logits $h = xW_g$, where $W_g \in \mathbb{R}^{d \times N}$. The computational complexity (FLOPs) for the projection is:

$$\mathcal{C}_{\text{std}} = 2 \cdot d \cdot N \tag{i}$$

The parameter storage requirement is $\mathcal{M}_{\text{std}} = d \cdot N$. With $N$ scaling to millions (e.g., $10^6$), both storage and computation become prohibitive (e.g., $2 \cdot 10^9$ FLOPs per token just for routing).

The proposed **Cartesian Product Router** decomposes the expert index space into a grid of size $N_r \times N_c$ (where $N = N_r \cdot N_c$). It employs two projection matrices $W_r \in \mathbb{R}^{d \times N_r}$ and $W_c \in \mathbb{R}^{d \times N_c}$. The logit computation involves two smaller projections: $s_r = xW_r$ and $s_c = xW_c$. The computational complexity becomes:

$$\mathcal{C}_{\text{cart}} = 2 \cdot d \cdot (N_r + N_c) \tag{ii}$$

Assuming a balanced grid where $N_r \approx N_c \approx \sqrt{N}$, the complexity is:

$$\mathcal{C}_{\text{cart}} \approx 4 \cdot d \cdot \sqrt{N} \tag{iii}$$

Comparing the two, the reduction factor is:

$$\frac{\mathcal{C}_{\text{std}}}{\mathcal{C}_{\text{cart}}} = \frac{2dN}{4d\sqrt{N}} = \frac{\sqrt{N}}{2} \tag{iv}$$

For $N = 10^6$, this yields a theoretical speedup of $500\times$ in router projection FLOPs. Similarly, the parameter storage scales with $O(\sqrt{N}d)$ instead of $O(Nd)$, making million-scale expert routing feasible.

**Top-$K$ Selection Complexity.** While the Cartesian Product Router efficiently reduces projection complexity, selecting the top-$K$ experts from $N$ scores remains a challenge. Standard approaches materialize the full score matrix and sort it globally, costing $O(N)$ memory traffic and $O(N \log N)$ or $O(N)$ operations. Our implementation utilizes a fused **Block-wise Merge Selection** kernel that avoids full score materialization. We partition the $N$ experts into blocks of size $B_{sel}$ (e.g., 4096). For each block, the kernel: (1) Computes scores on-the-fly (Complexity: $O(B_{sel})$). (2) Performs iterative max-reduction to find the local top-$K$ candidates (Complexity: $O(B_{sel} \cdot K)$). (3) Merges local candidates with the global top-$K$ buffer (Complexity: $O(K^2)$). Summing over $N/B_{sel}$ blocks, the total time complexity per token is:

$$\mathcal{C}_{\text{select}} = \frac{N}{B_{sel}} \cdot (B_{sel} \cdot K + K^2) = O(N \cdot K + \frac{N}{B_{sel}}K^2) \tag{v}$$

The $O(K^2)$ term arises from merging local candidates into the global buffer, typically implemented via insertion sort within GPU registers. Our fused approach eliminates the dominant $O(N)$ global memory I/O bottleneck. By keeping scores in registers/SRAM, the operation becomes compute-bound and effectively negligible in latency on modern GPUs.

**Memory Traffic Analysis for Scheduling.** We analyze the memory I/O volume for the routed FFN execution. Let $L$ be the number of tokens in a batch, and $K$ be the number of activated experts per token. The computation involves retrieving expert parameters $W, V \in \mathbb{R}^{N \times d}$ and performing the forward pass. We assume the worst-case scenario for baselines where no cache reuse occurs due to large $N$ and scattered access patterns.

**Token-Centric Scheduling (Baseline).** In this paradigm, each token $x_l$ retrieves parameters for its specific top-$K$ selected experts indices $\mathcal{I}_l = \{I_{l,0}, \ldots, I_{l,K-1}\}$. The total memory traffic for loading expert parameters is proportional to the total number of expert executions:

$$\mathcal{D}_{\text{token}} \propto \sum_{l=0}^{L-1} \sum_{k=0}^{K-1} (Size(W_{I_{l,k}}) + Size(V_{I_{l,k}})) = 2d \cdot L \cdot K \tag{vi}$$

This approach suffers from redundancy when multiple tokens select the same expert. Furthermore, the memory accesses are non-contiguous (gather operations), significantly degrading effective bandwidth utilization.

**Expert-Centric Scheduling (Ours).** This strategy inverts the loop order. We first identify the set of unique activated experts in the batch: $\mathbb{E}_{\text{active}} = \bigcup_{l=0}^{L-1} \mathcal{I}_l$. The execution groups all tokens assigned to a specific expert $E_j \in \mathbb{E}_{\text{active}}$. Consequently, the parameters for expert $E_j$ are loaded exactly once from HBM to SRAM. The total memory traffic for expert parameters becomes:

$$\mathcal{D}_{\text{expert}} \propto \sum_{j \in \mathbb{E}_{\text{active}}} \left( Size(W_j) + Size(V_j) \right) = 2d \cdot |\mathbb{E}_{\text{active}}| \tag{vii}$$

The reduction in memory traffic is defined by the ratio:

$$\eta = \frac{\mathcal{D}_{\text{token}}}{\mathcal{D}_{\text{expert}}} = \frac{L \cdot K}{|\mathbb{E}_{\text{active}}|} \tag{viii}$$

In our OmniMoE settings, where experts are fine-grained but $K$ is large, each expert is frequently accessed by multiple tokens in a batch (i.e., $L \cdot K \gg |\mathbb{E}_{\text{active}}|$). Thus $\eta \gg 1$, indicating a substantial reduction in parameter I/O.

**Scheduling Overhead and Token Traffic.** While expert-centric scheduling optimizes parameter loading, it introduces a reordering step and necessitates token reloading. We analyze these costs below:

*1. Sort Overhead:* The preprocessing involves flattening the routing indices and sorting $M = L \times K$ tasks by (Expert Group, Token ID). The complexity is $O(M \log M)$. Given that GPU memory bandwidth is the primary bottleneck, this lightweight integer sorting (performed efficiently via radix sort) is negligible. Empirically, scheduling occupies $< 5\%$ of total latency, well-amortized by the speedup in the GEMM phase.

*2. Token Memory Access:* A potential concern is that if a token activates experts across multiple groups, it must be loaded multiple times. However, our hierarchical sorting ensures that within each expert group, tokens are processed in *increasing order of Token ID*. This converts token access into a **strictly sequential stream**, enabling perfectly coalesced memory reads. Unlike the random access patterns in token-centric baselines, our approach fully utilizes the high sequential bandwidth of HBM. Furthermore, since $K$ is large, tasks for the same token often cluster in consecutive groups, allowing effectively cached reuse.

## B. Experimental Setup

In this section, we provide detailed configurations for the experiments conducted in Sec. 3. To ensure a rigorous evaluation, we prioritize **architectural comparison** via controlled pre-training from scratch. Rather than comparing against off-the-shelf checkpoints, which differ in training data and recipes, we represent baselines using their underlying architectural prototypes (e.g., Gshard, DeepSeekMoE) trained on a strictly identical corpus. This isolates the impact of the MoE architectural design from confounding factors. Although academic resource constraints limit pre-training to the 1.7B activated-parameter scale, we verify that all methods adhere to predictable scaling laws (Section 3.2), ensuring our findings extrapolate to larger scales.

Detailed implementation specifications for all evaluated methods are as follows: For coarse-grained baselines (Gshard, DeepSeekMoE), we adopt the best-performance kernels released by NVIDIA in CuTile. For fine-grained baselines (PKM, PEER), we employ highly-optimized Triton fused kernels. For OmniMoE, we implement our custom Expert-Centric Scheduling using Triton to maximize hardware utilization.

*Table A.* **Speed and Memory Benchmarking Configurations**. We list the key hyperparameters used for measuring inference latency and memory usage. Total Params denotes the total parameter count of the FFN layer, while Act Params refers to the number of parameters active during the forward pass of a single token. Act Tokens represents the number of tokens in a batch. $d$ is the hidden dimension size. $N$ is the total number of experts. $d_{rffn}$ and $d_{sffn}$ denote the intermediate dimensions of the routed FFN and shared FFN (if applicable), respectively. $K$ indicates the number of experts selected per token.

| ALGO | TOTAL PARAMS | ACT PARAMS | ACT TOKENS | $d$ | $N$ | $d_{rffn}$ | $d_{sffn}$ | $K$ |
|---|---|---|---|---|---|---|---|---|
| Gshard | 200M | [3M, 6M, 12M, 25M] | [1k, 2k, 4k, 8k, 16k] | 1024 | 128 | 512 | - | [2, 4, 8, 16] |
| DeepSeekMoE | 200M | [6M, 9M, 15M, 28M] | [1k, 2k, 4k, 8k, 16k] | 1024 | 256 | 256 | 1024 | [4, 8, 16, 32] |
| PKM | 200M | [3M, 5M, 13M, 26M] | [1k, 2k, 4k, 8k, 16k] | 1024 | 197136 | - | - | [2500, 5000, 12500, 25000] |
| PEER | 200M | [3M, 5M, 13M, 26M] | [1k, 2k, 4k, 8k, 16k] | 1024 | 102400 | - | - | [1250, 2500, 6250, 12500] |
| OmniMoE | 200M | [4M, 7M, 15M, 28M] | [1k, 2k, 4k, 8k, 16k] | 1024 | 102400 | - | 1024 | [512, 1024, 2048, 4096] |

*Table B*. **Language Modeling Configurations**. Detailed hyperparameters for pre-training experiments across different scales (Activation 80M, 200M, 680M, and 1.7B). Total Params and Act Params indicate the total model size and the per-token active parameter count, respectively. Number of Step and Number of Batch specify the training duration and total tokens seen. LR is the peak learning rate. $n_{layer}$ and $d_{model}$ denote the number of transformer layers and the hidden dimension. Tied Emb indicates whether the input and output embeddings are tied.

| ALGO | TOTAL PARAMS | ACT PARAMS | NUMBER OF STEP | BATCH TOKENS | LR | $n_{layer}$ | $d_{model}$ | TIED EMB |
|---|---|---|---|---|---|---|---|---|
| Activation 80M | | | | | | | | |
| Dense | 80M | 80M | 13500 | 0.128M tokens | 3e-3 | 12 | 768 | ✓ |
| Gshard | 280M | 80M | 13500 | 0.128M tokens | 3e-3 | 12 | 768 | ✓ |
| DeepSeekMoE | 280M | 80M | 13500 | 0.128M tokens | 3e-3 | 12 | 768 | ✓ |
| PKM | 280M | 80M | 13500 | 0.128M tokens | 3e-3 | 12 | 768 | ✓ |
| PEER | 280M | 80M | 13500 | 0.128M tokens | 3e-3 | 12 | 768 | ✓ |
| OmniMoE | 280M | 80M | 13500 | 0.128M tokens | 3e-3 | 12 | 768 | ✓ |
| Activation 200M | | | | | | | | |
| Dense | 200M | 200M | 20800 | 0.192M tokens | 2e-3 | 16 | 1024 | ✓ |
| Gshard | 800M | 200M | 20800 | 0.192M tokens | 2e-3 | 16 | 1024 | ✓ |
| DeepSeekMoE | 800M | 200M | 20800 | 0.192M tokens | 2e-3 | 16 | 1024 | ✓ |
| PKM | 800M | 200M | 20800 | 0.192M tokens | 2e-3 | 16 | 1024 | ✓ |
| PEER | 800M | 200M | 20800 | 0.192M tokens | 2e-3 | 16 | 1024 | ✓ |
| OmniMoE | 800M | 200M | 20800 | 0.192M tokens | 2e-3 | 16 | 1024 | ✓ |
| Activation 680M | | | | | | | | |
| Dense | 680M | 680M | 35000 | 0.392M tokens | 1e-3 | 24 | 1536 | ✓ |
| Gshard | 2.7B | 680M | 35000 | 0.392M tokens | 1e-3 | 24 | 1536 | ✓ |
| DeepSeekMoE | 2.7B | 680M | 35000 | 0.392M tokens | 1e-3 | 24 | 1536 | ✓ |
| PKM | 2.7B | 680M | 35000 | 0.392M tokens | 1e-3 | 24 | 1536 | ✓ |
| PEER | 2.7B | 680M | 35000 | 0.392M tokens | 1e-3 | 24 | 1536 | ✓ |
| OmniMoE | 2.7B | 680M | 35000 | 0.392M tokens | 1e-3 | 24 | 1536 | ✓ |
| Activation 1.7B | | | | | | | | |
| Dense | 1.7B | 1.7B | 40000 | 1M tokens | 1e-3 | 28 | 2048 | ✗ |
| Gshard | 6.4B | 1.7B | 40000 | 1M tokens | 1e-3 | 28 | 2048 | ✗ |
| DeepSeekMoE | 6.4B | 1.7B | 40000 | 1M tokens | 1e-3 | 28 | 2048 | ✗ |
| PKM | 6.4B | 1.7B | 40000 | 1M tokens | 1e-3 | 28 | 2048 | ✗ |
| PEER | 6.4B | 1.7B | 40000 | 1M tokens | 1e-3 | 28 | 2048 | ✗ |
| OmniMoE | 6.4B | 1.7B | 40000 | 1M tokens | 1e-3 | 28 | 2048 | ✗ |

Table A outlines the configurations used for **Speed and Memory Benchmarking**. In this benchmark, we evaluate system performance by varying one dimension while keeping the other fixed at its minimum value. Specifically, when sweeping the activated parameter budget (Act Params), the number of activated tokens (Act Tokens) is fixed at 1K. Conversely, when varying the number of activated tokens, the activated parameter budget is held at the minimum configuration for each method. Table B presents the comprehensive **Language Modeling Configurations** for different model scales. We detail the model architecture hyperparameters (total parameters, activated parameters, layer count, model dimension, etc.) and training recipes (learning rate, batch size, training steps) to ensure reproducibility.

## C. Expert Parallelism Communication Overhead

To verify the communication efficiency of OmniMoE in large-scale distributed training scenarios, we conducted distributed experiments focused on the Expert Parallelism (EP) communication overhead. We investigated the impact of the number of experts and sequence length on communication bandwidth.

**Scalability with Number of Experts.** Figure A illustrates the change in communication overhead as the number of experts scales from 1K to 2M under a fixed sequence length.

**Key Finding: Decoupling Communication Cost from Model Capacity.** The experimental results reveal a significant

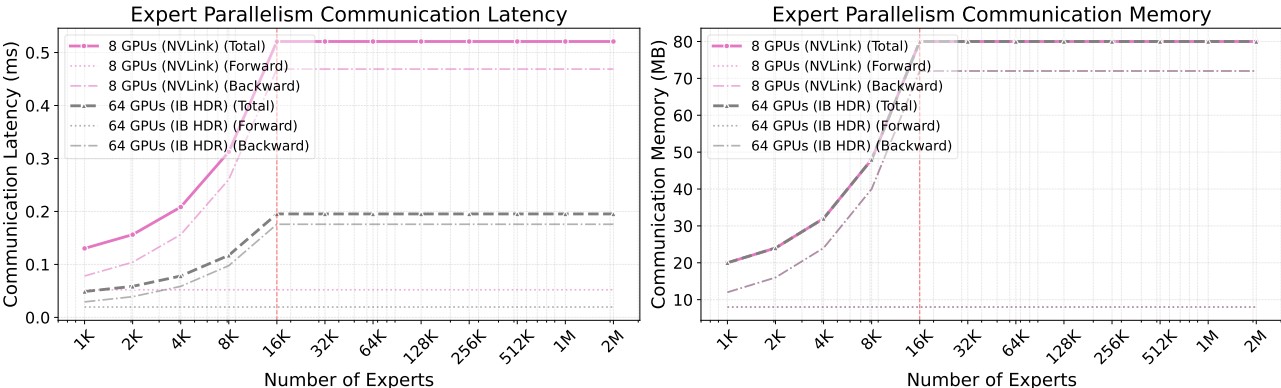

*Figure A.* **Communication overhead vs. Number of Experts.** The communication latency and memory usage saturates and remains constant as the number of experts increases beyond the activation count.

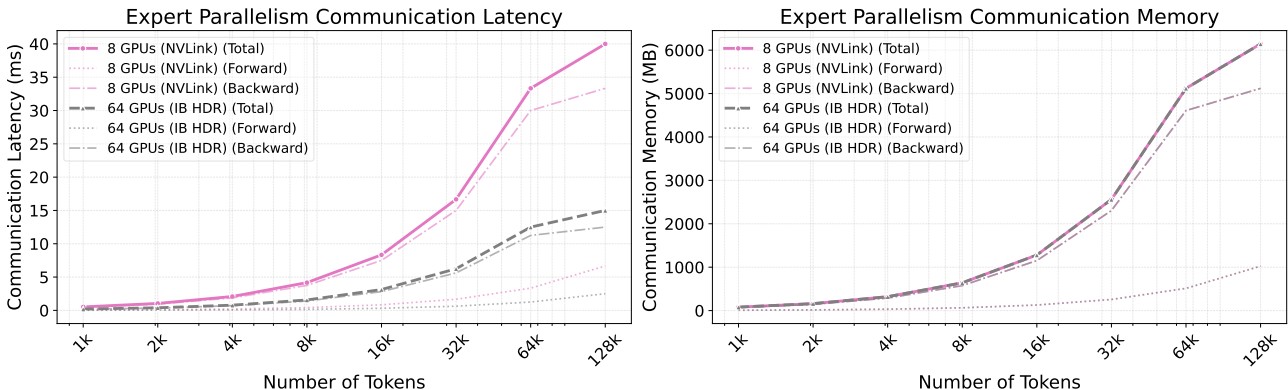

*Figure B.* **Communication overhead vs. Sequence Length.** Communication volume scales linearly with sequence length.

"saturation effect": when the number of experts $N$ exceeds the total number of activated experts ($n_{tokens} \times K = 16,384$), the total communication volume for the backward pass stabilizes at approximately 80MB and does not grow with increasing $N$. This implies that OmniMoE successfully breaks the bottleneck where communication overhead grows linearly with model capacity in traditional architectures. In an 8-GPU environment, this communication latency is merely 0.521 ms, which is negligible. This demonstrates OmniMoE's capability to support scaling to millions of experts with constant communication cost.

**Scalability with Sequence Length.** Figure B shows the communication overhead as the number of tokens scales from 1K to 128K under a fixed number of experts.

**Key Finding: Linear Communication Growth.** Since the essence of EP communication is token distribution and gradient aggregation, the communication volume exhibits a strictly linear relationship with the sequence length $n_{tokens}$. Even in extreme scenarios with ultra-long sequences, where the total communication volume is approximately 6GB, the estimated latency on a 64-GPU cluster is only 15 ms. This indicates that OmniMoE's communication mechanism remains efficient for long-sequence training and does not become a bottleneck for training throughput.

