# OpenReview forum: "OmniMoE: An Efficient MoE by Orchestrating Atomic Experts at Scale"
_ICML.cc/2026/Conference — ICML 2026 regular_

### Official Review · Reviewer_udNS · 2026-03-03

**Soundness:** 3
**Presentation:** 3
**Significance:** 2
**Originality:** 2
**Overall Recommendation:** 4
**Confidence:** 5

**Summary:**

This paper proposes OmniMoE, a system-algorithm co-designed Mixture-of-Experts framework that pushes expert granularity to vector-level "Atomic Experts.". As token level routing is computationally expensive, the paper introduces a Dynamic Expert Assembly (DEA) mechanism that retrieves and composes top-K atomic experts per token from a large selection pool. To make this practical, they first introduce (i) a Cartesian Product Router that reduces the N-dimension routing projection into two √N-dimensional subspaces, reducing routing complexity from O(N) to O(√N) and (ii) Expert-Centric Scheduling that groups experts to convert scattered memory accesses into batched Grouped GEMM operations. On seven zero-shot benchmarks at the 6.4B-A1.7B scale, OmniMoE achieves 50.9% average accuracy, outperforming methods like DeepSeekMoE and PEER, while speeding up inference efficiency by 10.9x compared to PEER.

**Compliance With Llm Reviewing Policy:**

Affirmed.

**Final Justification:**

My concerns have been answered, and I'd like to keep the score.

**Key Questions For Authors:**

1. The Cartesian Product Router factorizes row and column routing independently, yet this assumption is never empirically validated. Could you report the approximation error it introduces，for example, the KL divergence between the factorized joint distribution and the ground-truth from full dense router?
2. Expert-Centric Scheduling shares the same core principle as Klotski and ExpertFlow: batching tokens per expert to convert scattered memory accesses into contiguous, coalesced reads. Could you provide a controlled latency comparison against these systems and clarify the key technical differences?
3. The reported benchmarks focus on factual recall and common-sense association. Given that Dynamic Expert Assembly resembles a learned retrieval scheme, whether the accuracy gains hold on tasks requiring multi-step reasoning is also important. Could you include results on at least one reasoning benchmark (e.g., GSM8K or MATH), even at the 280M scale?.

**Limitations:**

Not adequately discussed. It would be helpful if the authors discussed the conditions under which the Cartesian Product Router's independence assumption may break down.

**Strengths And Weaknesses:**

**Strengths:**
1. The experiment is very solid, All baselines are trained on the same corpus with identical Transformer backbones at multiple scales (280M–6.4B), using all efficient kernel implementations for competitors.
2. The ablation study clearly reveals the effective nessess of proposed method of Cartesian Product Router to improve the model accuracy and Expert-Centric Scheduling for improving inference efficiency.
3. The paper makes effective use of bold text throughout, allowing readers to scan fluidly and locate key claims at a glance. Figure 1 gives a clear architectural overview that intuitively contrasts coarse-grained MoE, fine-grained MoE, and OmniMoE. Figure 2 offers a well-structured illustration of the Dynamic Expert Assembly mechanism, making the core technical contribution easy to follow.
4. The paper addresses a real and important gap between applying fine-grained routing precision and hardware efficiency, which is a key bottleneck for scaling MoE models. The 10.9× latency reduction over PEER is a striking practical result.
5. The Cartesian Product Router and Expert-Centric Scheduling are clean, well-motivated contributions that naturally extend product-key ideas (PKM) to the MoE routing setting. The system-algorithm co-design perspective is valuable.

**Weaknesses:**
1. The Cartesian Product Router rests on an assumption that row and column routing are independent, which is intuitive but left unvalidated. It would strengthen the work to quantify the approximation error this introduces.
2. The speed and memory benchmarking is conducted exclusively at 200M parameters, which falls outside the 280M–6.4B range used in the scaling-law experiments, leaving it unclear whether this is a deliberate design choice or an oversight. Given that the paper itself states all inference runs use a single node with 8×NVIDIA A100 GPUs, there seems to be no hardware barrier to evaluating at a more representative scale (e.g., 6.4B).
3. The evaluation suite covers only common-sense and factual recall tasks. Since OmniMoE's Dynamic Expert Assembly resembles a learned look-up-table retrieval scheme, it may naturally favour memorisation-heavy benchmarks over tasks requiring multi-step reasoning. Whether the gains hold on benchmarks such as GSM8K or MATH remains untested, and including even one reasoning task would considerably strengthen the generality argument.
4. Relative to PKM, the primary algorithmic novelty of the Cartesian Product Router lies in reducing the routing parameter to accelerate training convergence. As discussed under Soundness, the enforced row–column independence may compromise accurate expert selection.
5. The Expert-Centric Scheduling contribution has conceptual overlap with `Klotski: Efficient Mixture-of-Expert Inference via Expert-Aware Multi-Batch Pipeline` and `ExpertFlow: Optimized Expert Activation and Token Allocation for Efficient Mixture-of-Experts Inference`, both of which share the same core principle of aggregating tokens per expert to maximize memory reuse and GPU utilization. The paper includes neither a direct empirical comparison with these systems nor a substantive discussion of what technically distinguishes the proposed scheduling from these closely related prior works.

---

> ### Author Rebuttal · Authors · 2026-03-30
>
> ### **Response to Q1, W1, W4**:
> We thank the reviewer for those insightful questions. In principle, comparing the factorized router with a full dense router would be informative.
> However, at our scale, the dense router is not a meaningful reference: Table 2 (w/o Cartesian Product Router) shows **severe routing collapse**, with only 4% expert usage, much higher unevenness, which means much worse load balance,  and higher perplexity. We therefore assess factorization through routing quality and load balancing. As shown in Table 2, the **Cartesian Product Router** achieves lower perplexity, **100% expert usage and much lower unevenness** (0.24 vs. 0.77), where unevenness is the KL divergence between the empirical expert activation distribution and the uniform distribution. While this is not the exact KL divergence to a dense router, it is the more meaningful diagnostic in our setting because the unfactorized dense router itself collapses. Overall, these results suggest that Cartesian Product factorization is not only a scalable way to reduce routing cost, but also empirically yields more stable and balanced routing at massive scale.
>
> ### **Response to  Q2, W5**
> We thank the reviewer for this insightful connection. OmniMoE shares the same high-level goal as Klotski and ExpertFlow: improving hardware efficiency through expert-centric execution. However, the execution setting is fundamentally different. **Klotski and ExpertFlow are designed for coarse-grained MoEs** with a small number of pre-materialized full-FFN experts, whereas **OmniMoE operates over hundreds of thousands of vector-level atomic experts**.
>
> This difference makes the systems incomparable in a direct latency study. In Klotski and ExpertFlow, the expert-side weights are fixed during execution, and the main systems problem is to **batch and reorder the tokens assigned to each expert**. In OmniMoE, however, **the expert-side operands are also dynamic**: different tokens activate different atomic experts, so the system must not only batch tokens, but also **dynamically assemble token-dependent dense expert blocks before grouped GEMM**. This introduces two additional challenges beyond coarse-grained expert scheduling: scalable routing/load balancing over a massive expert space, and dynamic expert assembly for efficient execution.
>
> For this reason, a direct latency comparison with Klotski or ExpertFlow is not well-defined, since these systems are **built for token scheduling over fixed expert blocks** rather than **joint expert assembly and token batching**. We will clarify this distinction in the related-work section of the final manuscript if accepted.
>
>
> ### **Response to  Q3, W3**:
> We thank the reviewer for this constructive suggestion. In addition to the results in the main paper, we further evaluate OmniMoE at the **6.4B-A1.7B** scale on **GSM8K, MATH, BBH, MBPP, and HumanEval**, providing additional evidence of its effectiveness on mathematical reasoning, broad reasoning, and code generation benchmarks. On two multi-step reasoning benchmarks, the gains remain clear: **GSM8K:** OmniMoE 69.8, DeepSeekMoE 65.6, PEER 31.4; **MATH:** OmniMoE 24.1, DeepSeekMoE 21.3, PEER 7.9. Detailed evaluation results shows below.
>
> *Table I. Additional Results on Reasoning and Code Generation Benchmarks*
> | Model| GSM8K | MATH | BBH  | MBPP | HumanEval | Avg. |
> | - | - | ---- | ---- | ---- | --------- | ---- |
> | Dense  | 46.3  | 11.7 | 37.7 | 40.0 | 6.7       | 32.3 |
> | GShared| 61.9  | 18.9 | 41.5 | 47.4 | 8.5       | 38.6 |
> | DeepSeekMoE    | 65.6  | 21.3 | 43.0 | 49.9 | 9.1       | 40.5 |
> | PKM| 18.9  | 4.1  | 24.8 | 17.6 | 1.8       | 19.1 |
> | PEER| 31.4  | 7.9  | 35.6 | 29.7 | 4.9       | 27.0 |
> | OmniMoE (ours) | 69.8  | 24.1 | 44.6 | 52.3 | 9.8       | 42.5 |
>
> These results suggest that OmniMoE’s gains extend beyond factual recall and commonsense to multi-step reasoning. The large gap over PEER indicates that a purely fine-grained design is insufficient for sustained reasoning, while OmniMoE benefits from combining precise routed retrieval with a universally activated shared dense MLP. The consistent improvement over DeepSeekMoE further supports the effectiveness of the overall design.
>
> ### **Response to W2**:
> We thank the reviewer for pointing this out. The **200M** figure refers to the parameter count of a **single FFN/MoE layer**, not the full model. Our efficiency evaluation is therefore conducted as a **single-layer micro-benchmark**, which is designed to isolate the routing and scheduling overhead from the rest of the Transformer stack. This setting is also practically meaningful: with **28 layers**, a **200M** FFN/MoE layer corresponds to roughly **5.6B** FFN/MoE parameters in total, placing the benchmark in a realistic large-model regime.
>
> We agree that this was not stated clearly enough in the main text. If accepted, we will clarify in Section 3.1 that the **200M** benchmark is a **per-layer efficiency comparison**, to avoid this confusion.

---

> > ### Author Rebuttal · Reviewer_udNS · 2026-04-02
> >
> > Thanks for the rebuttal. Everything is clear now.

---

> > > ### Author Response · Authors · 2026-04-02
> > >
> > > Dear Reviewer udNS,
> > >
> > > Thank you very much for your acknowledgment and for the time you dedicated to reviewing our rebuttal. We are delighted to hear that our responses have fully addressed your concerns and that the clarifications were helpful.
> > >
> > > As we move toward the final stages of the discussion period, we would like to kindly ask if there is any further information we can provide to support a positive recommendation for our work. If you are satisfied with the clarifications and the improvements we have proposed, we would deeply appreciate it if you could reflect this in your final evaluation.
> > >
> > > Thank you again for your constructive feedback, which has significantly improved the quality of our work.
> > >
> > > Best regards,
> > >
> > > Authors

---

### Official Review · Reviewer_7eTp · 2026-03-08

**Soundness:** 2
**Presentation:** 3
**Significance:** 3
**Originality:** 3
**Overall Recommendation:** 4
**Confidence:** 2

**Summary:**

This paper presents OmniMoE, a system-algorithm co-designed Mixture-of-Experts framework that addresses the fundamental tension between expert granularity and hardware execution efficiency in MoE architectures. The authors identify three key bottlenecks in scaling fine-grained MoEs — limited expressivity, routing overhead, and memory inefficiency — and propose three tightly coupled solutions: Atomic Experts with Dynamic Expert Assembly (DEA), a Cartesian Product Router, and Expert-Centric Scheduling. Experiments on seven benchmarks demonstrate superior downstream accuracy and a 10.9× inference speedup over the state-of-the-art fine-grained baseline PEER.

**Compliance With Llm Reviewing Policy:**

Affirmed.

**Key Questions For Authors:**

1. Can you report results controlling for $K$ across methods (or a sweep of $K$) to isolate the architectural efficiency gains from the sheer number of experts activated?
2. How much of the 10.9× speedup over PEER is strictly due to the Expert-Centric Scheduling algorithm versus optimizations within your custom Triton kernel?

**Limitations:**

yes.

**Strengths And Weaknesses:**

Strength
1. The paper elegantly solves the tension between MoE granularity and hardware efficiency through a tightly integrated design: Atomic Experts, the Cartesian Product Router, and Expert-Centric Scheduling.
2. The reported 10.9× inference speedup over PEER is impressive.
3. Evaluating via controlled pre-training from scratch across multiple scales (up to 1.7B active parameters) eliminates confounding variables. The empirical work is well-supported by complexity analyses in the appendix.

Weakness
1. The Cartesian Router assumes row and column selections are independent ($p(i,j|x) \approx p_r(i|x)p_c(j|x)$). In reality, expert specialization likely induces strong correlations. The paper lacks empirical validation showing this assumption doesn't degrade routing quality.
2. The premise that the shared MLP handles general semantics while atomic experts specialize in long-tail knowledge is intuitive but unproven. There are no probing experiments or activation analyses to back this up.
3. Atomic experts are essentially rank-1 computations. The paper lacks a theoretical comparison showing that a weighted sum of $K$ rank-1 units can approximate the same function class as a standard MLP of equivalent size.
4. OmniMoE relies on custom Triton kernels, making it difficult to separate algorithmic speedups from low-level engineering.  A controlled comparison against baselines using a uniform implementation is missing.
5. OmniMoE uses $K=4096$, PEER uses $K=12500$, and DeepSeekMoE uses $K=32$. Because the scheduling speedup depends heavily on $K$, the lack of a controlled, matched-$K$ ablation obscures the true architectural advantage.

---

> ### Author Rebuttal · Authors · 2026-03-29
>
> ### **Response to W1**
> We thank the reviewer for this important point. Although the Cartesian Product Router imposes a factorized approximation, Table 2 suggests that it **does not harm routing quality in practice**. Replacing it with a standard router reduces expert usage to 4% and worsens perplexity by 1.4×, whereas our router **maintains 100% expert usage and low unevenness (0.24)**. OmniMoE also achieves the best zero-shot accuracy and perplexity among the compared variants, supporting that the factorized design preserves strong routing quality in our setting.
>
> ### **Response to W2**
> We thank the reviewer for this insightful comment. To address this concern, we conduct both **quantitative logit-contribution analysis** and **qualitative token-level branch analysis**, both supporting our functional-specialization hypothesis.
>
> **Quantitative analysis.** On the full MMLU test set, we divide target tokens into common and uncommon groups by empirical frequency and measure the relative logit contribution of the Shared Dense MLP and the atomic-expert branch. For **uncommon tokens**, the atomic-expert branch contributes **61.1%** of the total logit contribution; for **common tokens**, its contribution drops to **27.6%**, with the shared branch contributing the remaining 72.4%.
>
> *Table I. Logit Contribution Across Token Frequencies*
> |Branch|Common words|Uncommon words|
> | - | - | - |
> |DEA|0.276|0.611|
> |Shared|0.724|0.389|
>
> **Qualitative analysis.** We further analyze branch contributions on a specialized reasoning example (link: https://anonymous.4open.science/r/omnimoe-EAC2/token_case_study.pdf). Low-frequency, domain-specific terms such as *multicollinearity* and *causality* receive stronger contribution from the atomic-expert branch, while the shared dense branch contributes more to syntactic glue, structural phrasing, and common linguistic patterns.
>
> Together, these results support that the shared MLP mainly captures general linguistic structure, while the atomic-expert branch contributes more strongly to long-tail and specialized content.
>
> ### **Response to W3**
> We thank the reviewer for raising this point. OmniMoE’s routed branch is not simply a linear sum of independent rank-1 units. **Dynamic Expert Assembly (DEA)** retrieves the selected top-K atomic experts and assembles them into token-dependent matrices $w_x, v_x \in \mathbb{R}^{K \times d}$, so that
>
> $y_{\text{routed}} = (g_x \odot \sigma(x w_x^\top)) v_x$.
>
> Under a fixed routing pattern, this has the same algebraic form as a **width-K gated one-hidden-layer MLP**: $w_x$ and $v_x$ are the input/output projections, and $g_x$ acts as neuron-wise gates. Thus, the routed branch should not be viewed as a mere weighted sum of isolated rank-1 units; rather, **DEA assembles the selected atomic experts into a standard nonlinear hidden-layer computation** once the routing decision is made. We will make this equivalence clearer in the camera-ready version if accepted.
>
> ### **Response to W4, Q2**
> We thank the reviewer for highlighting the importance of implementation fairness. Here we response to W4 and Q2 jointly, as both concern whether the reported speedup reflects architectural gains or merely low-level engineering. As described in Appendix B, we evaluate all methods with strong state-of-the-art optimized implementations rather than unoptimized reference code, since practical LLM efficiency depends critically on hardware-aware execution and a single generic implementation would obscure architectural differences. In particular, we use NVIDIA CuTile kernels for coarse-grained baselines and fused Triton kernels for fine-grained baselines.
>
> As to Q2, Expert-Centric Scheduling and the fused kernel in OmniMoE are co-designed: the former creates dense grouped-GEMM workloads, and the latter executes them efficiently. Removing only our kernel would therefore be artificial, especially since PEER is also evaluated with an optimized kernel. Instead, Table 2 isolates the scheduling effect: with the architecture and $K$ fixed, disabling Expert-Centric Scheduling increases latency by 24.8×, indicating that the speedup mainly comes from the proposed co-design rather than kernel optimization alone.
>
> ### **Response to W5, Q1**
> We thank the reviewer for emphasizing controlled comparisons. Matching the number of activated experts ($K$) is not the most principled control, because **expert capacity differs substantially across architectures**—from a full FFN block in DeepSeekMoE to a vector-level expert in OmniMoE and PEER. Equalizing $K$ therefore leads to unbalanced per-token computation. We therefore control the activated parameter budget (up to ∼28M active parameters), which is a fairer measure of compute. We also conducted an additional matched-$K$ experiment (see https://anonymous.4open.science/r/omnimoe-EAC2/speed_memory_grid.pdf); with the same $K$, the coarse-grained baseline still shows higher latency and memory than PEER and OmniMoE, so the conclusion remains unchanged.

---

> > ### Author Rebuttal · Reviewer_7eTp · 2026-04-03
> >
> > Thanks for clarifying. I will maintain the score.

---

> > > ### Author Response · Authors · 2026-04-07
> > >
> > > Dear Reviewer 7eTp,
> > >
> > > Thank you very much for your feedback and for acknowledging that your concerns have been fully addressed.
> > >
> > > We sincerely appreciate your meticulous review and the constructive comments provided during both the initial review and the rebuttal phase. Your insights have been invaluable in enhancing the quality of our work.
> > >
> > > Thank you again for your time and professional support.
> > >
> > > Best regards,
> > > Authors

---

### Official Review · Reviewer_Xw96 · 2026-03-09

**Soundness:** 3
**Presentation:** 3
**Significance:** 2
**Originality:** 2
**Overall Recommendation:** 3
**Confidence:** 2

**Summary:**

This paper proposes OmniMoE, a hybrid MoE layer that combines a shared dense MLP with a routed bank of vector-level "atomic experts." To make extremely fine-grained routing practical, the paper introduces a Cartesian Product Router that factorizes routing over a 2D expert grid, and an Expert-Centric Scheduling strategy that reorganizes token-expert interactions into grouped GEMM-friendly blocks. Experiments based on a 40B-token pretraining setup and seven zero-shot benchmarks show modest quality improvements over both coarse-grained and fine-grained baselines at a fixed activated-parameter budget, together with substantially lower reported inference latency than PEER and DeepSeekMoE.

**Compliance With Llm Reviewing Policy:**

Affirmed.

**Key Questions For Authors:**

For Questions 1-3, my concern is specifically about the paper's mechanistic interpretation of the two-branch design. The current ablations support the claim that both branches are useful, but they do not yet directly establish the stronger narrative that the shared dense branch mainly handles general semantics/reasoning while the routed atomic experts mainly handle long-tail knowledge retrieval. Strong evidence on this point would improve my assessment of both soundness and significance; otherwise, I would recommend softening that interpretive claim.

1. Can you provide expert-usage statistics that directly test the claimed division of labor between the two branches? For example, are routed atomic experts activated more often on rare entities, factual tokens, or other long-tail patterns, while the shared dense branch dominates more common linguistic patterns? A positive result would make the mechanistic interpretation much more convincing.

2. Can you provide a clearer task-level breakdown for this claim? In particular, does removing or weakening the shared dense branch disproportionately hurt reasoning-oriented benchmarks, while weakening the routed branch disproportionately hurts factual or knowledge-intensive benchmarks? This would help determine whether the proposed functional specialization is actually supported by the data.

3. Can you include qualitative case studies or token-level analyses showing when the shared dense branch versus the routed atomic experts contribute most strongly to the final prediction? Such examples would help distinguish a demonstrated mechanism from a plausible but unverified intuition.

**Limitations:**

No. The paper does not adequately discuss limitations or potential negative impacts. It should explicitly discuss the independence assumption in the factorized router, sensitivity to grouping and scheduling hyperparameters, the additional system complexity required to realize the speedups, the missing evidence for training-time scalability, and the compute/energy cost of large-scale MoE experimentation.

**Strengths And Weaknesses:**

Soundness: The paper addresses a real technical bottleneck in fine-grained MoEs, and the overall system-algorithm co-design is coherent. The latency ablations are directionally supportive, especially for the expert-centric scheduling component. However, the technical formulation is not fully convincing as written. Equation (4) defines an atomic expert using one input vector and one output vector, but then states that the nonlinearity is instantiated with SwiGLU; standard SwiGLU requires a gated two-projection structure, so the exact parameterization is unclear. In addition, the claimed routing complexity reduction is somewhat overstated: factorization clearly reduces router parameter size and projection cost, but the paper itself notes that top-k search over the implicit grid still performs total work that scales with the full expert space. As a result, the current evidence is promising but still not fully sufficient to support the strongest version of the paper's central claims.

Presentation: The paper is well structured and generally clear. The progression from atomic experts to factorized routing and then to system scheduling is logical, and the figures do a good job of conveying the core intuition. I also found the overall narrative technically disciplined rather than merely high-level. My remaining concerns are mostly about precision rather than readability: some notation could be tightened, and a few claims around efficiency and component roles should be stated a bit more carefully. Overall, however, the paper presents its ideas in a solid and reasonably rigorous way.

Significance: The problem is important. If the paper's efficiency claims hold under fully fair implementations, enabling hardware-efficient fine-grained MoEs would be practically meaningful. The reported latency gains are large enough to be interesting. Still, the paper does not yet demonstrate a broad enough empirical impact to justify a stronger recommendation. The zero-shot quality gain over the strongest coarse-grained baseline is modest, the evaluation is limited to seven benchmarks, and the central narrative about separating "general semantics" from "long-tail knowledge retrieval" is only indirectly supported.

Originality: The work contains a reasonable combination of modeling and systems ideas, and packaging them into one MoE design is useful. However, the novelty appears moderate rather than strong. Shared dense experts, product-structured routing, and execution reordering/grouped GEMM are all closely related to prior ideas. The new atomic-expert formulation is the most distinctive part, but it is also close in spirit to dynamically retrieved rank-1 or low-rank parameterization, and the paper does not compare against equally parameterized low-rank alternatives that would isolate this contribution more clearly.

---

> ### Author Rebuttal · Authors · 2026-03-30
>
> ### **Response to concerns in "Strengths And Weaknesses"**
> We sincerely thank the reviewer for the careful reading and helpful feedback. We agree that these points should be stated more precisely. First, the mention of SwiGLU in Eq. (4) is a presentation error: each **atomic expert** in our implementation uses **Swish**, and we will correct this if the paper is accepted. At the same time, after Dynamic Expert Assembly, the retrieved atomic experts are composed into a token-conditioned **sparse FFN** that is algebraically analogous to a **SwiGLU-style hidden layer**, rather than a standard low-rank approximation. Second, we do not claim that factorization removes the full routing cost; rather, it reduces router parameter size and projection cost, while the remaining implicit-grid search still scales with the full expert space. Importantly, our ablations show that under large expert counts, the practical routing bottleneck lies primarily in the former rather than the latter, and that our router yields a substantial latency reduction in this regime. Overall, OmniMoE is best viewed as a dynamic sparse expert assembly mechanism, not a conventional low-rank parameterization.
>
> ### **Response to Q1**
> We thank the reviewer for this insightful suggestion. We first clarify that expert-usage statistics alone cannot reliably reveal the division of labor between the two branches: both are activated for every token, and the activated parameter budget is fixed by design. We therefore analyze **branch specialization** through the **relative logit contribution** of the MoE branch.
> Specifically, we conduct a statistical analysis on the full **MMLU test set**, partitioning target tokens into common and uncommon groups by empirical frequency. The aggregated results are shown below.
>
> *Table I. Logit Contribution Across Token Frequencies*
> | BRANCH | Common words | uncommon words |
> | ------ | ------------ | -------------- |
> | DEA    | 0.276        | 0.611          |
> | Shared | 0.724        | 0.389          |
>
> These results show a clear and systematic pattern: **the DEA branch contributes substantially less on common words (27.6%), but much more on uncommon words (61.1%)**. This supports our interpretation that the shared dense branch mainly provides stable support for common linguistic patterns, while the routed DEA branch contributes more strongly to less frequent, more specialized tokens.
>
> Overall, we believe this token-level contribution analysis is more informative than raw expert-usage statistics for answering the reviewer’s question about functional specialization.
>
> ### **Response to Q2**
> We sincerely thank the reviewer for this insightful suggestion. While training fully ablated 6.4B models from scratch is computationally prohibitive during the short rebuttal period, **Table 2 already provides informative task-level evidence.**
>
> By isolating the two branches, we observe a clear cross-over degradation pattern consistent with our specialization hypothesis:
>
> 1. **Weakening the Shared Branch (****`w/o Shared Dense MLP`****):** Removing the shared expert disproportionately damages reasoning capabilities. The Reasoning Performance drops significantly by **-0.21x** (down to 0.79x), whereas the Knowledge Performance is relatively more resilient, dropping by only **-0.09x** (down to 0.91x).
> 2. **Weakening the Routed Branch (****`w/o Cartesian Product Router`****):** As shown in Table 2, removing our router causes expert usage to collapse to only 4%, effectively leaving the routed atomic-expert pathway severely weakened, underutilized, and consequently poorly trained. In this case, **Knowledge Performance** drops much more sharply to **0.66×**, while **Reasoning Performance** drops to **0.79×**.
>
> Taken together, these results provide quantitative evidence that the two branches play **complementary** roles: weakening the routed branch hurts knowledge-oriented performance more, whereas weakening the shared branch hurts reasoning-oriented performance more.
>
> ### **Response to Q3**
> We sincerely thank the reviewer for this helpful suggestion. We conducted a token-level case study on a specialized zero-shot reasoning task (a labor economics regression problem) and visualized branch dominance for each generated token (anonymous link: https://anonymous.4open.science/r/omnimoe-EAC2/token_case_study.pdf).
>
> The pattern is consistent with our interpretation: **routed atomic experts** contribute more to **low-frequency, domain-specific terms** (e.g., *multicollinearity*, *variance inflation factors*), while the **shared dense branch** contributes more to **syntactic glue, structural phrasing, and general logical framing** (e.g., “should not be interpreted causally because ...”). Overall, this case study provides qualitative token-level evidence that the two branches play complementary roles, with the shared branch supporting general linguistic structure and DEA contributing more to long-tail knowledge and domain-specific reasoning.

---

### Official Review · Reviewer_d8Xv · 2026-03-10

**Soundness:** 3
**Presentation:** 3
**Significance:** 3
**Originality:** 3
**Overall Recommendation:** 4
**Confidence:** 5

**Summary:**

This work proposes OmniMoE,  a system-algorithm co-designed MoE framework that integrates a shared dense MLP for general-purpose reasoning with massive atomic experts for long-tail knowledge retrieval, thereby enabling more precise parameter activation. It yields 10.9× inference speedup over the state-of-the-art fine-grained baseline.

**Compliance With Llm Reviewing Policy:**

Affirmed.

**Final Justification:**

My questions have been partially answered. Thus I'd like to increase my score.

**Key Questions For Authors:**

See weakness.

**Limitations:**

yes

**Strengths And Weaknesses:**

Strengths:
1. This work is derived from an important motivation - the tension between fine-grained MoE layers and hardware efficiency.

2. Evaluation setup and results in terms of efficiency and pretrained base model performance are convincing.

3. This work is overall well-presented with clear figures and tables.

Weakness:
1. The major concern is the small scale of experimental settings. 8x A100 is probably a little bit "out-of-date" to pretrain a MoE foundation model today, which has limited bandwidth and could amplify the weakness of fine-grained experts. It would be interesting to see how the comparison may be conducted on the latest industrial hardware (e.g. GB300 NVL72). I understand this may not be very feasible for authors to have during rebuttal, thus some theoretical analysis/insights would be appreciated.

2. It would be great to see how the performance/efficiency scales while changing the fine-granularity.

3. It would be very helpful to show more speedup wall-clock time results, e.g. time to first tokens and tokens per seconds

4. It's not very clear to me how the atomic expert works and what its express capability is. As it's a very light-weight non-linear vector unit, how does the combination of this "memory lookup" take the role of "reasoning", or does it? It would be very helpful for the authors to further illustrate.


If the authors are willing to address my questions (I understand some of them may be harsh), I'd be happy to increase my score. :=)

---

> ### Author Rebuttal · Authors · 2026-03-29
>
> ### **Response to W1**
> We thank the reviewer for this important suggestion. Due to rebuttal-time resource constraints, our experiments are limited to a single **8×A100** node. While we cannot provide new GB300/NVL72 results during rebuttal, we can offer a qualitative analysis based on current hardware scaling trends. Using **BF16** as a reference, according to NVIDIA A100 datasheet and the NVIDIA Blackwell Ultra Datasheet, **A100 80GB** provides about **312 TFLOPS** and **1.935 TB/s** HBM bandwidth, whereas **GB300/NVL72** provides about **2.5 PFLOPS** and **8 TB/s** HBM bandwidth per GPU. Thus, compute scales by roughly **8×**, while bandwidth scales by only about **4×**, implying a substantially higher compute-to-bandwidth ratio on newer hardware.
>
> This trend makes **memory efficiency** increasingly important for fine-grained MoEs, since conventional token-centric routing often leads to scattered, non-coalesced expert accesses and poor locality. **OmniMoE** is explicitly designed to mitigate this bottleneck: our **Expert-Centric Scheduling** improves locality and parameter reuse by grouping active experts and reorganizing irregular sparse accesses into denser **Grouped GEMM**-style execution, effectively **shifting the bottleneck more toward compute**. Therefore, newer hardware is unlikely to materially weaken OmniMoE’s advantage. If accepted, we will add this discussion to the final version.
> ### **Response to W2**
> We thank the reviewer for this valuable suggestion. An ablation over expert granularity would indeed be informative. OmniMoE focuses on the **vector-level Atomic Expert** because this setting **maximizes activation precision** and **minimizes redundant parameter usage**.
>
> More generally, changing granularity introduces a clear trade-off: **coarser experts** may simplify routing and scheduling, but they also **re-introduce redundant activation** and **reduce the precision of token-specific parameter composition**. Moreover, our current **system-algorithm co-design is specifically tailored to atomic experts**: the fused kernels, memory layout, and batching strategy in **Expert-Centric Scheduling** are built around the atomic-expert access pattern. Coarsening the expert granularity would therefore create a **system-level mismatch**, require **substantial kernel re-design**, and likely reduce the efficiency gains of the current implementation.
>
> In this submission, we prioritized validating the central claim that this **extreme fine-grained regime can still be made efficient** through joint algorithm-system co-design. If accepted, we will clarify this design choice and the associated granularity trade-off in the final manuscript.
>
> ### **Response to W3**
> We thank the reviewer for this helpful suggestion. We agree that end-to-end wall-clock efficiency is important for assessing practical deployment.
>
> For decoding, we report **throughput (tokens per second)**. Measured on 8×A100 GPUs with data parallelism, using 40 batched parallel requests with approximately 512 input tokens and 8192 output tokens each, **OmniMoE** achieves **14.2k tok/s**, compared with **11.6k tok/s** for **PEER** and **13.4k tok/s** for **DeepSeekMoE**.
>
> *Table I. Decoding Throughput Comparison*
> |Model|Throughput|
> |-|-|
> |Dense| ~14.8k tok/s |
> |GShared  | ~14.0k tok/s |
> |DeepSeekMoE| ~13.4k tok/s |
> |PKM | ~7.9k tok/s  |
> |PEER | ~11.6k tok/s |
> |OmniMoE (ours) | ~14.2k tok/s |
>
> For the first-token stage, although we did not separately report TTFT, we have measured **prefill latency** in our ablation study. These results suggest that both the proposed **Cartesian Product Router** and **Expert-Centric Scheduling** provide clear speedups during prefilling.
>
> Taken together, these results provide evidence that OmniMoE improves wall-clock efficiency in both the prefill and decode stages. If accepted, we will make this distinction clearer and present these benchmarks more explicitly in the final manuscript.
>
> ### **Response to W4**
> We appreciate the reviewer for this profound question. While a single atomic expert is lightweight, OmniMoE derives its expressive power from their **dynamic composition**, not from any individual vector unit.
>
> 1. **Expressivity via assembly.** Through **Dynamic Expert Assembly (DEA)**, the router retrieves multiple atomic experts and composes them into a token-conditioned $K \times d$ sparse FFN. Thus, the main expressive capacity comes from the assembled nonlinear computation, rather than from any single atomic expert alone.
> 2. **Empirical evidence.** Our ablation shows that even without the shared dense MLP, the routed atomic experts still retain **79%** of the model’s reasoning performance, indicating that they contribute substantially to reasoning-related capability.
> 3. **Interpretation.** Our current evidence supports a complementary role between the two branches, rather than a strict mechanistic separation. If accepted, we will clarify this complementary relationship in more detail in the final version.

---

> > ### Author Rebuttal · Reviewer_d8Xv · 2026-04-02
> >
> > Thank the authors for their time and effort. My questions have been partially answered. Thus I'd like to increase my score.

---

> > > ### Author Response · Authors · 2026-04-02
> > >
> > > Dear Reviewer d8Xv,
> > >
> > > Thank you very much for your feedback and positive reassessment of our work. We are encouraged by your recognition of our efforts and the clarifications provided during the rebuttal.
> > >
> > > We sincerely appreciate the time you invested in reviewing our research. Your constructive insights are very helpful for the further refinement of this work.
> > >
> > > Thank you again for your support.
> > >
> > > Best regards,
> > >
> > > Authors

---

### Decision · Program_Chairs · 2026-04-30

**Decision:**

Accept (regular)

**Comment:**

There was consensus amongst the reviewers that the work is well-motivated in seeking to address the tradeoff between increasing fine-grained specialization of experts in MoE models and hardware throughput. The reviewers' primary concerns pre-rebuttal included the relevance of the real-world timings to contemporary GPU hardware, the lack of evidence for the specialization assumption amongst the lower level atomic experts proposed, the expert independence assumption of the cartesian router proposed, the fairness of comparing across MoE models with different numbers of K activated experts.

Post-rebuttal the reviewers universally agreed that all or most of their concerns were well-addressed by the author's rebuttal and recommended weak accept, with the exception of Reviewer Xw96. Since, however, Reviewer Xw96 never completed the mandatory rebuttal acknowledgement or otherwise acknowledged the author's rebuttal, I believe it's prudent to discount Reviewer Xw96's remaining objection.